# Execution-based Code Generation using Deep Reinforcement Learning

**Parshin Shojaee**      *parshinshojaee@vt.edu*
**Aneesh Jain**      *aneeshj@vt.edu*
**Sindhu Tipirneni**      *sindhut@vt.edu*
**Chandan K Reddy**      *reddy@cs.vt.edu*
*Department of Computer Science, Virginia Tech, Arlington, VA*

**Reviewed on OpenReview:** *https://openreview.net/forum?id=0XBuaxqEcG*

## Abstract

The utilization of programming language (PL) models, pre-trained on large-scale code corpora, as a means of automating software engineering processes has demonstrated considerable potential in streamlining various code generation tasks such as code completion, code translation, and program synthesis. However, current approaches mainly rely on supervised fine-tuning objectives borrowed from text generation, neglecting unique sequence-level characteristics of code, including but not limited to compilability as well as syntactic and functional correctness. To address this limitation, we propose PPOCoder, a new framework for code generation that synergistically combines pre-trained PL models with Proximal Policy Optimization (PPO) which is a widely used deep reinforcement learning technique. By utilizing non-differentiable feedback from code execution and structure alignment, PPOCoder seamlessly integrates external code-specific knowledge into the model optimization process. It is important to note that PPOCoder is a task-agnostic and model-agnostic framework that can be used across different code generation tasks and PLs. Extensive experiments on three code generation tasks demonstrate the effectiveness of our proposed approach compared to SOTA methods, achieving significant improvements in compilation success rates and functional correctness across different PLs. The source code for PPOCoder can be found at `https://github.com/reddy-lab-code-research/PPOCoder`.

## 1 Introduction

Recent years have seen a surge of attention towards the use of deep learning and neural language models to automate code generation and other software engineering processes, as a means to enhance developer productivity. The software development process encompasses a variety of code generation tasks, including code completion (Code2Code) (Li et al., 2018), code translation (Code2Code) (Zhu et al., 2022b), and program synthesis (NL2Code) (Li et al., 2022). Inspired by the great performance of pre-trained neural language models (LMs) in different natural language processing (NLP) tasks, these pre-training techniques have been recently employed on large-scale code corpuses to automate code generation tasks. Examples of such pre-trained models include CodeBERT (Feng et al., 2020), CodeGPT (Lu et al., 2021), PLBART (Ahmad et al., 2021), and CodeT5 (Wang et al., 2021). However, the code generation domain brings its unique challenges. While it is essential for code to be human-readable, it is imperative that the generated code maintains syntactic and functional correctness, i.e., being able to pass the compilation and unit tests.

Despite the advancements of pre-trained code models, they are heavily influenced by NLP's self-supervised masked language modeling (MLM) and often struggle to ensure the execution correctness of the generated codes. For example, recent works (Chen et al., 2021a; Li et al., 2022; Jha & Reddy, 2023) have shown that a significant number of programs generated from pre-trained PL models using simple sampling methods may not pass the unit tests. To improve code generation towards operational accuracy, several approaches

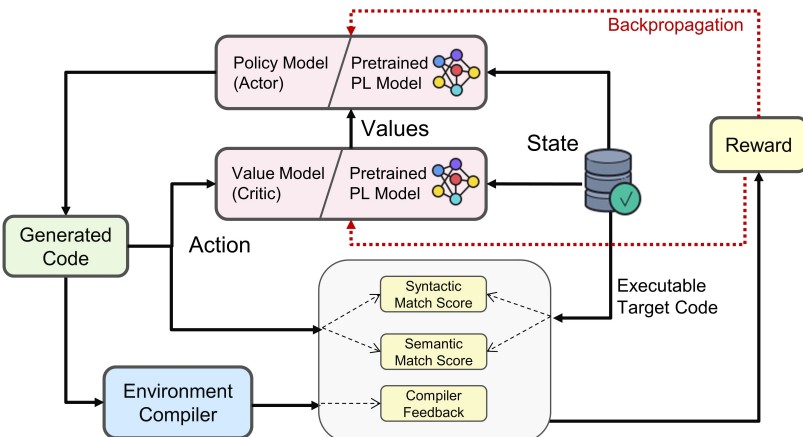

Figure 1: An overview of the proposed PPOCoder framework. The actor and critic networks are first initialized from the pre-trained PL model for the desired task. Following the sampling of a synthetic program from the stochastic policy, the reward is determined using the execution feedback and the ground truth target code. The values are estimated by the critic network. Finally, both actor and critic networks are updated based on the obtained values and returns.

are followed: (*i*) filtering and repairing the non-executable synthesized programs (Kulal et al., 2019), (*ii*) using energy-based generation models with execution constraints (Korbak et al., 2021), and (*iii*) using reinforcement learning (RL) fine-tuning mechanisms (Wang et al., 2022; Zhong et al., 2017; Le et al., 2022). However, existing approaches are often tailored to a specific programming language (PL) or task, e.g., (Le et al., 2022) is exclusively designed for program synthesis task in Python, and (Roziere et al., 2020) only targets code translation tasks in Python, Java, and C++ PLs. We propose **PPOCoder,** illustrated in Fig. 1, an RL framework for code generation that incorporates non-differentiable feedback derived from code execution and structure alignment as external, code-specific knowledge into the model optimization. PPOCoder utilizes the PPO (Schulman et al., 2017) algorithm for RL optimization which is based on the proximal actor-critic advantage policy gradient objective and a trust region mechanism, making the model optimization more stable and less sensitive to new environments (tasks, PLs, or datasets). Also, PPOCoder integrates discrete compiler feedback with the syntactic and semantic matching scores between the generated codes and executable targets. This integration reduces the sparsity of the reward function, leading to a better guidance of the policy to generate higher-quality code. *To the best of our knowledge, our work is the first one to integrate pre-trained PL models with PPO and incorporate non-differentiable execution and code structure elements into the feedback.* Inspired by (Ouyang al., 2022), PPOCoder also incorporates the KL-divergence penalty into the reward function to control explorations and prevent catastrophic forgetting, i.e., drastic deviations from the priors already acquired by the pre-trained model. Previous works (Le et al., 2022) have employed token-level matching objectives (i.e., cross-entropy loss) in the fine-tuning stage as well as pre-training of PL models to constrain these drastic deviations; however, overoptimizing for token-level matching frequently results in memorization and restricted performance when faced with new tasks and datasets. We observe that incorporating the KL-divergence penalty instead of token-matching objectives during fine-tuning can effectively minimize the likelihood of memorization (check zero-shot results of program synthesis in Table 4), fostering a more controlled and efficient exploration that generalizes adeptly to diverse environments. To summarize, the major contributions of this paper are as follows:

- We present a PPO-based RL framework for code generation, PPOCoder, that utilizes non-differentiable code-specific feedback as the external source of knowledge in model optimization. PPOCoder provides a more stable and generalizable yet accurate model optimization that is less sensitive to new environments (tasks, PLs, or datasets) and generate higher-quality codes.

- We develop a new code-specific reward function based on the discrete compiler feedback (compilation or unit test signal when available) received at the end of the generation episode as well as the syntactic and

semantic matching scores between the AST sub-trees and DFG edges of the sampled generations and the correct targets.

- We reduce the chance of memorization by incorporating a KL-divergence penalty into reward instead of a cross-entropy loss used in earlier works during the fine-tuning phase to control explorations and prevent deviations from the pre-trained priors.
- We demonstrate the effectiveness of PPOCoder through an extensive set of experiments across diverse code generation tasks (code completion, code translation, program synthesis) and PLs (C++, Java, Python, C#, PHP, C). PPOCoder outperforms the SOTA baselines, improving the compilation rate and functional correctness over different PLs. We also investigate the benefits of PPOCoder's reward elements and PPO optimization through ablation study.

The organization of the remainder of this paper is as follows: In Section 2, existing code generation methods utilizing pre-trained models, structure-based approaches, and RL methods for sequence generation are summarized. Section 3 delves into the specifics of our proposed PPOCoder method, including its various components. The experimental evaluation of our method on three code generation tasks: code completion, code translation, and program synthesis tasks, as well as the ablation study and case study, can be found in Section 4. Finally, the paper concludes in Section 5.

## 2 Related Work

### 2.1 Pre-trained Models for Code Generation

Recent research has focused on using pre-trained language models to automate code generation tasks leveraging large-scale pre-training on the extensive code corpus data available in open-source repositories (Lu et al., 2021; Zan et al., 2022; Niu et al., 2022). Notable examples of these pre-trained models include: (*i*) CodeBERT (Feng et al., 2020), which conducts encoder-only pre-training using Masked Language Modeling (MLM) and Replaced Token Detection tasks; (*ii*) CodeGPT (Lu et al., 2021), a comparable decoder-only GPT-based pre-trained model introduced alongside the CodeXGLUE benchmark; (*iii*) PLBART (Ahmad et al., 2021), an encoder-decoder transformer model pre-trained employing the denoising autoencoding (DAE) objective; and (*iv*) CodeT5 (Wang et al., 2021), another encoder-decoder pre-trained model subsequently proposed, built upon the T5 (Raffel et al., 2020) architecture and trained with code data in eight PLs. However, these pre-trained PL models tend to rely heavily on self-supervised objectives for text generation, while grappling with maintaining essential sequence-level attributes of code such as syntactic and operational accuracy in the generations.

### 2.2 Leveraging Structure in Code Generation

Lately, there has been a surge of interest in combining PL models with logical constructs such as abstract syntax trees (ASTs) (Kim et al., 2021; Rabinovich et al., 2017; Wang & Li, 2021), code sketches (Nye et al., 2019), and data-flow graphs (DFGs) (Yasunaga & Liang, 2020; Guo et al., 2021). For example, GraphCode-BERT (Guo et al., 2021) uses DFGs to incorporate semantic information into the encoded representations, but its decoder is unaware of the code structures. StructCoder (Tipirneni et al., 2022) presents a pre-trained structure-aware encoder-decoder architecture, where both the encoder and decoder components are aware of syntax and semantic relations. However, despite these efforts, many of these structure-aware code generation models still struggle to ensure the syntactic and operational accuracy of the generated codes. The primary reason is the misalignment between code-specific evaluation metrics and model optimization objectives. In other words, these models are not optimized for the non-differentiable code-specific objectives such as compilability, readability, or passing test cases. Consequently, this results in performance deficiencies.

### 2.3 RL for Sequence Generation

RL has been used to optimize non-differentiable metrics in sequence generation tasks (Ranzato et al., 2016; Bahdanau et al., 2017), such as using the REINFORCE (Williams, 1992) algorithm to improve BLEU (Papineni et al., 2002a) and ROUGE (Lin, 2004) scores in the translation and summarization models. Recently, InstructGPT (Ouyang et al., 2022) has introduced a new Reinforcement with Human Feedback (RLHF) fine-tuning procedure which shows that language models fine-tuned with RL can better follow non-differentiable human feedback. Unlike natural language tokens, generated code should follow some unique

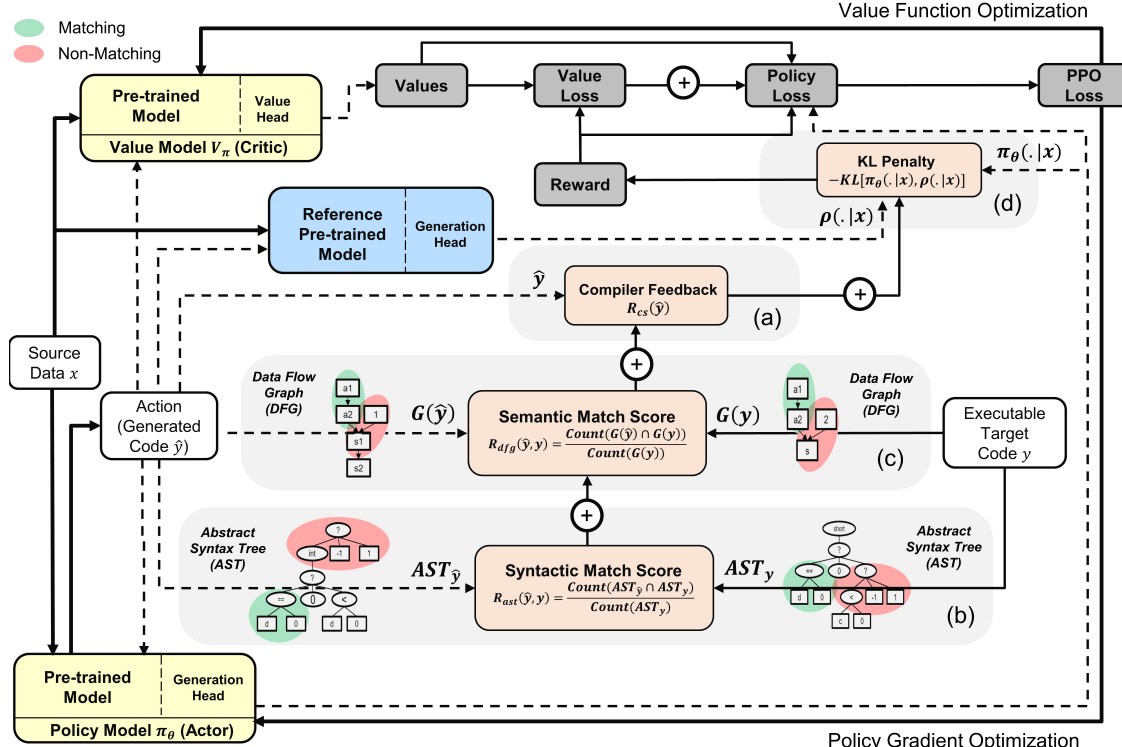

Figure 2: Overview of the PPOCoder with actor and critic models. The action is sampled from the policy based on the given source data $x$ (NL or PL). Then, a reward is obtained for each action to guide and control policy updates. The reward function is composed of four elements: ($a$) compiler feedback; ($b$) syntactic matching score based on ASTs; ($c$) semantic matching score based on DFGs; and ($d$) KL-divergence penalty between active policy and the reference pre-trained model. The critic model estimates value based on the obtained reward and PPOCoder will be optimized with PPO, which takes into account both value and policy optimization.

code-related attributes. This includes not only syntactic but also functional correctness, as the generated code must pass compilation and unit tests for machine execution. Recently, execution-guided approaches (Chen et al., 2019; Ellis et al., 2019; Chen et al., 2021b) and RL-based fine-tuning mechanisms (Wang et al., 2022; Zhong et al., 2017; Le et al., 2022) are used to enhance the quality of generated codes by PL models. For example, (Le et al., 2022) has recently studied the integration of RL with unit test signals in the fine-tuning of the program synthesis models. However, existing RL-based methods still encounter several limitations. They are often designed for a particular task (e.g., only program synthesis) or a particular PL (e.g., only Python), receive a sparse and discrete compiler signal only at the end of the generation episode, and are susceptible to memorization and poor performance on unseen data due to the use of token-matching loss in the RL fine-tuning step to prevent drastic deviations from the pre-trained PL model. Our model, PPOCoder, is designed to be both task- and model-agnostic, allowing the RL framework to be flexible for a wide range of code generation tasks and PLs. This is achieved by incorporating a PPO-based framework that combines compiler feedback with code structure elements in the reward function to address the sparsity, and employs a KL-divergence penalty to minimize large deviations, while reducing the chance of model memorization.

## 3   PPOCoder

PPOCoder provides a systematic mechanism for fine-tuning code generation models using deep reinforcement learning (RL) by incorporating compiler feedback and structure alignments as extra knowledge into the model optimization, thereby enhancing the quality of the generated codes in terms of code-specific sequence-level features such as syntactic and execution correctness. Fig. 2 shows the overall structure of our proposed PPOCoder model with the policy network (actor) $\pi_\theta$ responsible for code generation actions and the value

function (critic) $V_\pi$ responsible for the return estimations. They are both learned with the proximal policy optimization (PPO) approach taking reward $\mathcal{R}$. As shown in Fig. 2, the total reward is composed of four elements: ($i$) compiler feedback; ($ii$) syntactic match score; ($iii$) semantic match score; and ($iv$) KL-divergence penalty. We will provide further details about each of these components later in the paper.

### 3.1 Problem Formulation

The code generation procedure can be formulated as a sequential discrete finite-horizon Markov Decision Process (MDP) with the use of RL in which an agent interacts with the compiler over discrete horizon $T$ which is equivalent to the maximum number of generated code tokens. The proposed PPOCoder is formulated as follows:

**State $\mathcal{S}$:** The state of environment at each time-step, denoted as $s_t = (\hat{y}_{<t}, x), s_t \in \mathcal{S}$, is determined by the source PL/NL data $x$, as well as the set of generated code tokens before $t$, $\hat{y}_{<t}$.

**Action $\mathcal{A}$:** The PL model chooses the action at each time-step, denoted as $a_t = \hat{y}_t, a_t \in \mathcal{A}$, which is equivalent to the generated token at time-step $t$.

**Policy $\pi_\theta(a_t|s_t)$:** The stochastic policy network parameterized by $\theta$ is the downstream code generation model that predicts the next token conditioned on the previously generated tokens and the source data, so, $\pi_\theta(\hat{y}_t|\hat{y}_{<t}, x) : \mathcal{S} \to \Delta(\mathcal{A})$ where $\Delta(\mathcal{A})$ denotes the probability distribution over all actions (e.g., target vocabulary). The next action $\hat{y}_t$ will be decided based on the *top-k* sampling from this probability distribution. Policy is initialized with the pre-trained reference PL model $\rho$, i.e., $\pi_\theta^0(\cdot) = \rho$.

**Reward $\mathcal{R}$:** The reward $\mathcal{R}(\hat{y}, x, y)$ will be obtained at the end of the generation episode (i.e., after generating the $\langle endoftokens \rangle$ token) based on the generated code's syntactic and functional correctness as well as its structural alignment with executable codes. The reward function $\mathcal{R}(\cdot)$ is composed of different components which are explained in Section 3.2.

**Advantage $\hat{A}_\pi^t$:** Inspired by the Generalized Advantage Estimator (GAE) (Schulman et al., 2016), the advantage at time-step $t$ is defined as

$$\hat{A}_\pi^t = \delta_t + \gamma\delta_{t+1} + \ldots + \gamma^{T-t+1}\delta_{T-1},$$
$$\delta_t = r_t - V_\pi(\hat{y}_{<t}, x) + \gamma V_\pi(\hat{y}_{<t+1}, x), \tag{1}$$

where $\gamma$ is the discount rate; $r_t$ is the reward at time-step $t$; and $V_\pi(s_t)$ is the state value function at $t$ which can be approximated by a dense token-level value head on top of the hidden states of PL model.

**Objective:** The objective of PPOCoder is to find a policy that maximizes the expected reward of generated codes sampled from the policy.

$$\max_\theta \mathbb{E}_{x \sim \mathcal{X}, \hat{y} \sim \pi_\theta(.|x)}\big[\mathcal{R}(\hat{y}, x, y)\big], \tag{2}$$

where $\mathcal{X}$ is the training set of source data; $\pi_\theta(\cdot)$ is the policy network; and $\mathcal{R}(\cdot)$ is the reward function. We formulate the objective function as a maximization of the advantage instead of reward, as shown in Eq. (3), in order to reduce the variability of predictions.

$$\max_\theta \mathbb{E}_{x \sim \mathcal{X}, \hat{y} \sim \pi_\theta(.|x)} \left[\sum_{t=0}^{T} \hat{A}_\pi^t\big((\hat{y}_{<t}, x), \hat{y}_t\big)\right], \tag{3}$$

We adopt the policy gradient to estimate the gradient of non-differentiable reward-based objectives in Eqs. (2) and (3). Therefore, updating policy parameters for a given source data $x$ can be derived as

$$\max_\theta \mathcal{L}_\theta^{PG} = \max_\theta \mathbb{E}_{\hat{y} \sim \pi_\theta} \left[\sum_{t=0}^{T} \Big(log\pi_\theta(\hat{y}_t|\hat{y}_{<t}, x)\ \hat{A}_\pi^t\Big)\right], \tag{4}$$

$$\text{where}\ \ \nabla_\theta \mathcal{L}_\theta^{PG} = \mathbb{E}_{\hat{y} \sim \pi_\theta} \left[\sum_{t=1}^{T} \Big(\nabla_\theta log\pi_\theta(\hat{y}_t|\hat{y}_{<t}, x)\ \hat{A}_\pi^t\Big)\right],$$

where $\nabla_\theta \mathcal{L}_\theta^{PG}$ refers to the estimated gradient of objective function based on the policy parameterized by $\theta$. In order to further reduce the variations and avoid significantly changing the policy at each iteration, the objective function in Eq. (4) will be reformulated as shown in Eq. (5), called the conservative policy iteration (CPI).

$$\mathcal{L}_\theta^{CPI} = \mathbb{E}_{\hat{y} \sim \pi_\theta} \left[ \sum_{t=0}^{T} \left( \frac{log\pi_\theta(\hat{y}_t|\hat{y}_{<t}, x)}{log\pi_{\theta_{old}}(\hat{y}_t|\hat{y}_{<t}, x)} \ \hat{A}_\pi^t \right) \right] = \mathbb{E}_{\hat{y} \sim \pi_\theta} \left[ \sum_{t=0}^{T} \left( c_\pi^t(\theta) \ \hat{A}_\pi^t \right) \right], \tag{5}$$

where $\theta_{old}$ is the policy parameters before the update; and $c_\pi^t(\theta)$ is the ratio of log-probabilities from new and old policies.

## 3.2 Reward Function

Figure 2 illustrates that the reward of PPOCoder is composed of four different components which are designed to guide and control actions simultaneously towards generating higher-quality codes. These components are designed due to (1) the sparsity of compiler feedback which is only received at the end of code generation episode; and (2) the high chance of policy divergence from the pre-trained PL models. Check the effectiveness of these reward components in the reward ablation results of Section 4.4. Eq. (6) shows the combination of these different reward terms in the final reward vector $\mathcal{R}(\hat{y}, x, y) \in \mathbb{R}^T$ with $T$ as the generation episode length.

$$\mathcal{R}(\hat{y}, x, y) = (r_t : t = 1, \ldots, T), \tag{6}$$
$$r_t = \mathbb{1}(cond) \Big[ R_{cs}(\hat{y}) + R_{ast}(\hat{y}, y) + R_{dfg}(\hat{y}, y) - \beta R_{kl}(x, \hat{y}_{<t}) \Big]$$
$$+ \mathbb{1}(\neg cond) \left[ -\beta R_{kl}(x, \hat{y}_{<t}) \right]), \quad cond = (\hat{y}_t == \langle endoftokens \rangle)$$

where $r_t$ is the combined reward at time-step $t$; $R_{cs}(\cdot)$, $R_{ast}(\cdot)$, and $R_{dfg}(\cdot)$ are the compiler signal, syntactic match score, and the semantic match score reward terms, respectively. Note that these terms will be received at the end of the generation episode where $\hat{y}_t == \langle endoftokens \rangle$. The $R_{kl}(x, \hat{y}_{<t})$ is a KL-divergence penalty between the reference pre-trained priors and the active policy which is imposed to the reward function at each time-step to control the corresponding action. $\beta$ is also the coefficient of penalty to balance the combination of different reward terms.

### 3.2.1 Compiler Signal

For each source data $x$, we sample multiple generated codes in the target language based on the current policy network, $\hat{y} \sim \pi_\theta(.|x)$. Then, we pass these sampled codes $\hat{y}$ to a compiler and determine the reward based on the parsing signal. In case unit tests are available for the source data, the reward is determined by the functional correctness of generated codes, i.e., passing all unit tests, as shown in Eq. (7). If unit tests are not provided, compiler returns the syntactic correctness of generated codes as shown in Eq. (8). This reward term is designed to provide guidance to the model, encouraging it to take actions that result in the generation of higher-quality code in terms of syntactic/functional correctness.

Functional Correctness:
$$R_{cs}(\hat{y}) = \begin{cases} +1, & \text{if } \hat{y} \text{ passed all unit tests} \\ -0.3, & \text{if } \hat{y} \text{ failed any unit test} \\ -0.6, & \text{if } \hat{y} \text{ received RunTime error} \\ -1, & \text{if } \hat{y} \text{ received Compile error} \end{cases} \tag{7}$$

Syntactic Correctness:
$$R_{cs}(\hat{y}) = \begin{cases} +1, & \text{if } \hat{y} \text{ passed compilation test} \\ -1, & \text{otherwise} \end{cases} \tag{8}$$

### 3.2.2 Syntactic Matching Score

Since the compiler signal alone is too sparse, we also add additional information to better control and guide the structure of policy samples. To do so, we define a syntactic matching score $R_{ast}(\hat{y}, y)$ between the generated hypothesis $\hat{y} \sim \pi_\theta(.|x)$ and the parallel executable target $y$. The goal is to maximize this matching score for better compilability or syntactic correctness. We use the abstract syntax tree (AST) to find a tree representation of the code's abstract syntax structure. Then, we compare the sub-trees extracted from the hypothesis and reference target ASTs, respectively, and calculate the syntactic match score as a percentage of matched AST sub-trees.

$$R_{ast}(\hat{y}, y) = Count(AST_{\hat{y}} \cap AST_y)/Count(AST_y), \tag{9}$$

where $Count(AST_{\hat{y}} \cap AST_y)$ is the number of matched AST sub-trees between the hypothesis $\hat{y}$ and reference $y$; and $Count(AST_y)$ is the total number of reference AST sub-trees. While simple token-based comparison can detect some syntactic errors like missing tokens, it might fail to identify the context of these tokens and their structural relationship with each other in the code. AST-based analysis can provide more information by recognizing and comparing the structures of the code samples beyond just individual tokens. For instance, a missing operator or erroneous type casting—issues often undetected at the token level—would result in different AST sub-trees, flagging these as errors. Therefore, this score takes into account the structural correctness of the entire code syntax which is a richer and more informative measure of syntactic correctness. More details of implementation are provided in Appendix A.

### 3.2.3 Semantic Matching Score

To improve the functional correctness, we need to also take into account the semantic matching between hypothesis $\hat{y}$ and the executable target $y$, in addition to their syntactic matching. In PLs, code semantics are closely related to the dependencies of its variables. As a result, in order to construct a semantic matching score, we make use of the data-flow graphs (DFGs), a graph representation of code in which the nodes stand in for variables and the edges for the sources of each variable's values. Following (Guo et al., 2021), we denote DFG of a code $Y$ as $\mathcal{G}(Y) = (V; E)$ where $V = \{v_1, \ldots, v_m\}$ is the set of variables, and $e_{i,j} = \langle v_i, v_j \rangle$ is the $i \rightarrow j$ edge showing that value of the $j$-th variable originates from the $i$-th variable. Then, we calculate the semantic match score as a percentage of matched data-flows in DFGs.

$$R_{dfg}(\hat{y}, y) = Count(\mathcal{G}(\hat{y}) \cap \mathcal{G}(y))/Count(\mathcal{G}(y)), \tag{10}$$

where $Count(\mathcal{G}(\hat{y}) \cap \mathcal{G}(y))$ represents the number of matched DFG edges between hypothesis $\hat{y}$ and reference $y$; and $Count(\mathcal{G}(y))$ represents the total number of reference DFG edges. Maximizing this score can guide and control policy to generate codes which are more aligned with executable target codes in terms of variable relations, thus, enhancing the semantic quality and logical correctness of the generated codes. More details of implementation are provided in Appendix A.

### 3.2.4 KL-Divergence Constraint

We incorporate a negative KL-divergence penalty $KL(\pi||\rho)$ into the reward to prevent the active policy $\pi$ deviating away from the pre-trained PL model $\rho$. The KL-penalty at time $t$ can be approximated as:

$$R_{kl}(x, \hat{y}_{<t}) = KL(\pi||\rho) \approx \log \frac{\pi(.|x, \hat{y}_{<t})}{\rho(.|x, \hat{y}_{<t})} = \log(\pi(\cdot|x, \hat{y}_{<t})) - \log(\rho(\cdot|x, \hat{y}_{<t})), \tag{11}$$

where $\log(\pi(\cdot|x, \hat{y}_{<t}))$ and $log(\rho(\cdot|x, \hat{y}_{<t}))$ are the log-probabilities obtained from the active policy $\pi$ and pre-trained model $\rho$ at time $t$ given source data $x$ and the previously predicted tokens $\hat{y}_{<t}$. This reward term can control actions and play the role of entropy bonus in controlling exploration and exploitation where greater $\beta$ in Eq. (6) provides less exploration and more exploitation.

### 3.3 Loss Function

We employ proximal policy optimization (PPO) (Schulman et al., 2017) and define the loss function of PPOCoder as follows.

$$\mathcal{L}_\theta = -\mathcal{L}_\theta^{CPI} + \alpha\mathcal{L}_\theta^{VF}, \tag{12}$$

$$\mathcal{L}_\theta^{CPI} = \mathbb{E}_{\hat{y} \sim \pi_\theta}\left[\sum_{t=0}^{T} \min\left(c_\pi^t(\theta)\hat{A}_\pi^t, clip\left(c_\pi^t(\theta), 1 - \epsilon, 1 + \epsilon\right)\hat{A}_\pi^t\right)\right], \tag{13}$$

$$\mathcal{L}_\theta^{VF} = \mathbb{E}_{\hat{y} \sim \pi_\theta}\left[\sum_{t=0}^{T}\left(V_\pi(\hat{y}_{<t}, x) - \left(\hat{A}_\pi^t + V_{\pi_{old}}(\hat{y}_{<t}, x)\right)\right)^2\right]. \tag{14}$$

where the loss function $\mathcal{L}_\theta$ is the linear combination of surrogate policy objective function $\mathcal{L}_\theta^{CPI}$ and the value function squared error term $\mathcal{L}_\theta^{VF}$. Therefore, minimizing the loss function leads to the maximization

of the surrogate advantage policy objective (actor optimization) as well as the minimization of value error (critic optimization). In other words, the actor is guided to maximize the advantage policy objective which is correlated with maximizing the expected reward as explained in Eqs. (4)-(5); and the critic is enforced to minimize the token-level value estimation error which is defined based on the difference between the values of new policy $V_\pi(\hat{y}_{<t})$ and the estimated dense returns of the old policy $\hat{A}_\pi^t + V_{\pi_{old}}(\hat{y}_{<t})$. In Eqs. (12)-(14), $\epsilon$ is the proximal policy ratio clip range, and $\alpha$ is the linear combination weight between loss terms of actor and critic.

Alg. 1 provides the pseudocode of PPOCoder. For each source-target pair $(x, y)$, we sample multiple hypotheses from the policy network $\hat{y} \sim \pi_\theta(.|x)$. After generating each hypothesis, we find the integrated reward based on the reward function defined in Section 3.2, estimate the advantage, calculate the corresponding PPO loss function, and update the policy and value head parameters based on the final gradients (as shown in lines 5-21).

---

**Algorithm 1:** PPOCoder

**Input:** Set of parallel source-target input data pairs $(\mathcal{X}, \mathcal{Y})$, Pre-trained PL model $\rho$
**Output:** Fine-tuned policy with parameter $\theta$ based on RL

1  Initialize policy $\pi_\theta \leftarrow \rho$
2  **for** *number of epochs until convergence* **do**
3      **for** $(x, y) \in (\mathcal{X}, \mathcal{Y})$ **do**
4          **repeat**
5              $\hat{y} \leftarrow \pi_\theta(.|x)$
6              # Calculate Reward
7              Compute $R_{cs}(\hat{y})$ using Eq. (7) or Eq. (8)
8              Compute $R_{ast}(\hat{y}, y)$ using Eq. (9)
9              Compute $R_{dfg}(\hat{y}, y)$ using Eq. (10)
10             Compute $R_{kl}(x, \hat{y}_{<t})$ using Eq. (11)
11             $\mathcal{R}(\hat{y}, x, y) \leftarrow (r_t : t = 1, \ldots, T)$ where

$$r_t = \mathbb{1}(cond)\left[R_{cs}(\hat{y}) + R_{ast}(\hat{y}, y) + R_{dfg}(\hat{y}, y) - \beta R_{kl}(x, \hat{y}_{<t})\right]$$

13             $+\mathbb{1}(\neg cond)\left[-\beta R_{kl}(x, \hat{y}_{<t})\right])$; using Eq. (6)
14             # Estimate Advantage
15             $\hat{A}_\pi^t \leftarrow r_t - V_\pi(\hat{y}_{<t}, x) + \gamma V_\pi(\hat{y}_{<t+1}, x)$; using Eq. (1)
16             # Calculate Loss

17
$$\mathcal{L}_\theta^{CPI} \leftarrow \mathbb{E}_{\hat{y} \sim \pi_\theta}\left[\sum_{t=0}^{T} \min\left(c_\pi^t(\theta)\hat{A}_\pi^t, clip\left(c_\pi^t(\theta), 1-\epsilon, 1+\epsilon\right)\hat{A}_\pi^t\right)\right]$$

18
$$\mathcal{L}_\theta^{VF} \leftarrow \mathbb{E}_{\hat{y} \sim \pi_\theta}\left[\sum_{t=0}^{T}\left(V_\pi(\hat{y}_{<t}, x) - \left(\hat{A}_\pi^t + V_{\pi_{old}}(\hat{y}_{<t}, x)\right)\right)^2\right]$$

19             $\mathcal{L}_\theta \leftarrow \alpha\mathcal{L}_\theta^{VF} - \mathcal{L}_\theta^{CPI}$
20             # Update Model Parameters
21             $\theta \leftarrow \theta - \nabla_\theta \mathcal{L}_\theta$
22         **until** *num_samples*
23     **end**
24 **end**

---

## 4 Experiments

We evaluate PPOCoder on three different code generation tasks: (*i*) *Code Completion* automatically completes partial Python code snippets; (*ii*) *Code Translation* involves translating between any language-pair among six different PLs (Python, Java, C#, C++, PHP, C); and (*iii*) *Program Synthesis* (NL2Code) generates a Python function given a natural language (NL) description.

### 4.1 Code Completion

For this downstream task, we employ the Python corpus in CodeSearchNet (CSN) [1] (Husain et al., 2019). We extract 50K compilable Python methods with sufficient length (at least 64 tokens) and randomly split the data to train/val/test sets with 40K/5K/5K samples. We mask the last 25 tokens of the source code and ask the model to complete it. To evaluate the quality of generated codes, three metrics are used: (*i*) *Exact Match* (xMatch) which checks if the prediction is the same as the ground truth, (*ii*) *Levenshtein Edit Similarity*

---

[1] https://github.com/github/CodeSearchNet#data-details

(Edit Sim) (Lu et al., 2021; Svyatkovskiy et al., 2020) which is based on the number of single-character edits needed to match the generated code with the correct target, and (*iii*) *Compilation Rate* (Comp Rate) (Kulal et al., 2019) that shows the success rate of compilation among completed programs. Since unit tests are not provided, we focus on the syntactic correctness of the completed codes and take the compiler signal as reward.

Table 1 shows the results of PPOCoder along with the baselines on the code completion task. In this table, the BiLSTM (Luong et al., 2015) and Transformer (Vaswani et al., 2017) models are not pre-trained. The GPT-2 (Radford et al., 2019) model was pre-trained on text corpus, while CodeGPT (Lu et al., 2021) and CodeT5 (Wang et al., 2021) models are pre-trained on the large-scale source code corpus. The reported results for these pre-trained models are after the fine-tuning step on the code completion task. More details of the experimental setup are provided in Appendix A. It can be observed that CodeGPT and CodeT5 have a compilation rate of 46.84% and 52.14%, respectively, indicating that about half of the generated codes are not compilable. By employing our proposed PPOCoder framework on the fine-tuned CodeT5 model (PPOCoder + CodeT5), the compilation rate improves significantly from 52.14% to 97.68%, demonstrating the importance of incorporating compiler and structure alignment feedback into the model's optimization and the effectiveness of PPOCoder in code completion. We can also see that the PPOCoder performs similarly to other baselines in terms of Edit sim and xMatch scores, showing that the actor model effectively explores without deviating much from the pre-trained priors. In other words, the higher compilation rate achieved by PPOCoder +CodeT5 is not due to generating simple, arbitrary codes that are easy to compile.

Table 1: Results on the code completion task for completing the last 25 masked tokens from CodeSearchNet.

| Model | ↑xMatch | ↑Edit Sim | ↑Comp Rate |
|---|---|---|---|
| BiLSTM | 20.74 | 55.32 | 36.34 |
| Transformer | 38.91 | 61.47 | 40.22 |
| GPT-2 | 40.13 | 63.02 | 43.26 |
| CodeGPT | 41.98 | 64.47 | 46.84 |
| CodeT5 (220M) | 42.61 | 68.54 | 52.14 |
| PPOCoder + CodeT5 (220M) | **42.63** | **69.22** | **97.68** |

## 4.2 Code Translation

We use the XLCoST [2] (Zhu et al., 2022a) dataset for the code translation task which is a parallel dataset that includes solutions for problems related to data structures and algorithms in six languages: C++, Java, Python, PHP, C, and C#. In our experiments, we only use the compilable filtered parallel data in source and target language pairs. Table 6 in Appendix B shows the detailed statistics of these compilable filtered samples across all six PLs. To evaluate the quality of translated codes, we use two metrics: (*i*) *Comp Rate* that measures compilation success rate, and (*i*) *CodeBLEU* (Ren et al., 2020) score which combines the weighted BLEU (Papineni et al., 2002b) based on the code-related keywords with the syntactic and semantic alignment measures. As unit tests are not available for parallel language pairs, we focus on syntactic correctness with the help of compiler signal.

Table 2 presents the results of PPOCoder on XLCoST code translation tasks along with the baselines. In this table, column and row headers represent the translation source and target PLs, respectively. The Naive Copy baseline (Lu et al., 2021) simply copies the source code as the output, showing how similar two PLs are. The reported results of pre-trained CodeBERT and PLBART are after fine-tuning on the XLCoST code translation task for each language pair. More details of the experimental setup and implementation are provided in Appendix A. Table 2 demonstrates that incorporating our proposed PPOCoder +CodeT5 improves the overall *Comp Rate* across all language pairs, in comparison to the baseline CodeT5. Specifically, we observe an absolute increase of 9.92%, 22.22%, 21.62%, 13.20%, 7.46%, and 6.11% in the compilation rate for C++, Java, Python, C#, PHP, and C target PLs, respectively. In addition to PPOCoder's considerable improvement in the syntactic correctness of generated codes for different target PLs, PPOCoder +CodeT5 also obtains a comparable and mostly better CodeBLEU score to other baselines, meaning that PPOCoder +CodeT5's generated codes maintain the semantic meaning and often align better with the correct targets. This demonstrates that the surge in compilation rate does not derive from generating arbitrary short and

---

[2]`https://github.com/reddy-lab-code-research/XLCoST`

Table 2: Performance comparison of PPOCoder and baselines on XLCoST. The column and row language headers represent the translation source and target languages. The "Overall" column shows the weighted average scores over six PLs. The best results are shown in **bold** font.

| | Model | High Resource | | | | | | | | Low Resource | | | | Overall | |
|---|---|---|---|---|---|---|---|---|---|---|---|---|---|---|---|
| | | C++ | | Java | | Python | | C# | | PHP | | C | | | |
| | | ↑*CodeBLEU* | ↑*CompRate* | ↑*CodeBLEU* | ↑*CompRate* | ↑*CodeBLEU* | ↑*CompRate* | ↑*CodeBLEU* | ↑*CompRate* | ↑*CodeBLEU* | ↑*CompRate* | ↑*CodeBLEU* | ↑*CompRate* | ↑*CodeBLEU* | ↑*CompRate* |
| **C++** | Naive Copy | – | – | 44.56 | 20.28 | 17.81 | 9.73 | 47.28 | 21.25 | 19.83 | 8.21 | 63.94 | 4.62 | 38.68 | 12.82 |
| | CodeBERT | – | – | 62.56 | 37.12 | 36.41 | 26.72 | 67.12 | 38.52 | 38.77 | 12.23 | 21.84 | 2.31 | 45.34 | 23.38 |
| | PLBART | – | – | 71.23 | 44.51 | 69.09 | 45.92 | **74.74** | 51.86 | 62.35 | 53.63 | 52.76 | 36.22 | 66.03 | 46.42 |
| | CodeT5 | – | – | 80.17 | 59.01 | 72.83 | 53.33 | 73.11 | 60.31 | 67.47 | 68.21 | **66.02** | 71.44 | 71.92 | 62.46 |
| | PPOCoder + CodeT5 | – | – | **81.14** | **70.33** | **74.03** | **63.35** | 72.93 | **69.18** | **68.24** | **80.02** | 64.21 | **79.03** | **72.11** | **72.38** |
| **Java** | Naive Copy | 52.32 | 14.50 | – | – | 36.51 | 22.16 | 69.04 | 41.05 | 39.91 | 2.10 | 54.18 | 2.10 | 50.39 | 16.38 |
| | CodeBERT | 69.21 | 30.21 | – | – | 44.51 | 43.51 | 74.86 | 55.01 | 48.33 | 10.72 | 19.53 | 0 | 51.28 | 27.89 |
| | PLBART | 72.41 | 47.12 | – | – | 70.31 | 53.79 | 76.19 | 45.75 | 64.06 | 21.47 | 46.21 | 7.22 | 65.23 | 35.67 |
| | CodeT5 | 78.52 | 59.81 | – | – | 75.98 | 60.61 | 83.14 | 70.66 | 63.54 | 64.67 | **64.71** | 67.89 | 73.18 | 64.73 |
| | PPOCoder + CodeT5 | **79.14** | **82.80** | – | – | **76.65** | **92.14** | **85.66** | **86.80** | **64.16** | **90.88** | 60.52 | **82.16** | **73.22** | **86.95** |
| **Python** | Naive Copy | 37.41 | 21.47 | 39.72 | 17.27 | – | – | 38.52 | 10.71 | 43.91 | 16.84 | 35.11 | 0 | 38.93 | 13.26 |
| | CodeBERT | 68.93 | 42.15 | 45.76 | 38.10 | – | – | 40.23 | 26.10 | 52.12 | 31.74 | 18.32 | 0 | 45.07 | 27.62 |
| | PLBART | 74.49 | 61.20 | 63.82 | 54.59 | – | – | 67.35 | 44.65 | 69.86 | 66.76 | 39.15 | 6.12 | 62.93 | 46.66 |
| | CodeT5 | 79.86 | 74.11 | 74.15 | 62.74 | – | – | 75.54 | 58.26 | **79.83** | 80.05 | **56.83** | 70.81 | **73.24** | 69.19 |
| | PPOCoder + CodeT5 | **80.34** | **88.72** | **75.12** | **92.70** | – | – | **76.09** | **83.33** | 79.65 | **93.51** | 52.15 | **95.80** | 72.67 | **90.81** |
| **C#** | Naive Copy | 44.51 | 10.74 | 71.61 | 13.14 | 40.09 | 0 | – | – | 37.79 | 2.41 | 60.17 | 4.52 | 50.83 | 6.16 |
| | CodeBERT | 74.51 | 18.02 | 81.25 | 27.88 | 50.83 | 3.75 | – | – | 58.64 | 6.85 | 22.93 | 0 | 57.63 | 11.30 |
| | PLBART | 78.38 | 36.25 | 80.73 | 57.19 | 69.43 | 6.65 | – | – | 70.12 | 48.40 | 54.36 | 8.00 | 70.61 | 31.29 |
| | CodeT5 | 81.49 | 53.87 | 84.78 | 69.73 | **71.23** | 56.81 | – | – | 71.46 | 75.12 | 67.53 | 62.00 | 75.29 | 63.51 |
| | PPOCoder + CodeT5 | **82.94** | **68.51** | **85.77** | **81.92** | 70.43 | **78.61** | – | – | **72.06** | **82.62** | **68.11** | **71.90** | **75.86** | **76.71** |
| **PHP** | Naive Copy | 26.33 | 6.12 | 25.61 | 10.23 | 34.66 | 16.10 | 26.87 | 6.41 | – | – | 35.95 | 0 | 29.88 | 7.77 |
| | CodeBERT | 50.26 | 11.62 | 46.81 | 13.48 | 56.72 | 32.86 | 50.43 | 13.21 | – | – | 28.45 | 2.20 | 46.53 | 14.67 |
| | PLBART | 74.43 | 80.47 | 70.22 | 61.96 | 75.21 | **86.50** | 69.17 | 75.32 | – | – | 56.23 | 0 | 69.05 | 60.85 |
| | CodeT5 | 83.43 | 84.80 | 80.09 | 84.12 | **85.62** | 78.12 | 81.79 | 83.20 | – | – | 65.14 | 61.52 | 79.21 | 78.35 |
| | PPOCoder + CodeT5 | **85.55** | **89.50** | **82.12** | **90.31** | 83.26 | 82.52 | **83.88** | **89.80** | – | – | 65.01 | **76.92** | **79.96** | **85.81** |
| **C** | Naive Copy | 66.41 | 10.71 | 59.12 | 0 | 40.27 | 0 | 59.83 | 2.10 | 43.54 | 0 | – | – | 53.83 | 2.56 |
| | CodeBERT | 22.72 | 6.80 | 21.19 | 0 | 21.34 | 0 | 31.52 | 3.50 | 21.71 | 0 | – | – | 23.69 | 12.06 |
| | PLBART | 68.45 | 25.52 | 38.56 | 24.10 | 34.53 | 6.12 | 49.51 | 26.08 | 45.17 | 0 | – | – | 47.24 | 16.36 |
| | CodeT5 | 79.18 | 46.40 | 74.12 | 42.80 | **66.31** | 44.60 | **73.21** | 41.32 | 64.28 | 38.42 | | – | **71.42** | 42.71 |
| | PPOCoder + CodeT5 | **82.17** | 46.40 | **74.30** | **53.52** | 62.15 | **50.14** | 71.09 | **51.72** | **64.37** | **42.32** | – | – | 70.92 | **48.82** |

easy-to-compile codes. This is due to the KL-divergence penalty in the reward function that controls explorations and ensures the policy does not deviate too far from the pre-trained priors. Check Appendix F for examples of PPOCoder's translated codes. Also, among high-resource languages in Table 2, results show relatively greater compilation rate improvements for Python and Java as target PL. This is likely due to their high-level constructs, such as the absence of pointers and memory management constructs, which can be a source of errors in languages like C++ and C. Additionally, Java and Python feature a more lenient compilation process and extensive runtime error checking, resulting in many errors that would cause C++ compilation to fail, being detected only at runtime. The table shows a significantly lower compilation rate for code translation with C as target PL among all baselines. This is likely due to the limited number of samples with C as a target PL in the XLCoST dataset which is not sufficient for model fine-tuning (as shown in Table 6 of Appendix B).

We also evaluated the performance of PPOCoder on TransCoder [3] code translation benchmark (Roziere et al., 2020) with unit tests. Check Appendix C for the details.

### 4.3 Program Synthesis

In this task, we use the APPS (Hendrycks et al., 2021) dataset comprising 10k coding problems of varying difficulty levels, split 50/50 for train/test sets. The dataset consists of Introductory, Interview, and Competition level problems with respective train/test samples of 2639/1000, 2000/3000, and 361/1000. Each problem has 23 Python solutions and 21 unit tests on average. To evaluate the generated codes, we employ the *pass@k* metric, following (Chen et al., 2021a), which calculates the percentage of problems for which all unit tests are passed using $k$ synthetically generated program samples per problem. Since unit tests are provided in APPS, we use them in the PPOCoder's reward (as defined in Eq. (7)).

Table 3 demonstrates the results of program synthesis on the APPS dataset along with other baselines reported in (Hendrycks et al., 2021) including GPT-2 (Radford et al., 2019), GPT-3 (Brown et al., 2020), GPT-Neo (Black et al.), Codex (Chen et al., 2021a), AlphaCode (Li et al., 2022) and CodeRL (Le et al., 2022). The reported results for various models are post-fine-tuning on APPS, except for GPT-3 and Codex. For the experimental setup details of all methods, please refer to Appendix A. The results indicate that the smaller

---

[3] https://github.com/facebookresearch/TransCoder

Table 3: Results of the program synthesis task on the APPS dataset. The "All" column shows the weighted average scores over three difficulty levels. The best results are shown in **bold** font.

| Model | Size | ↑pass@1 | | | | ↑pass@5 | | | | ↑pass@1000 | | | |
|---|---|---|---|---|---|---|---|---|---|---|---|---|---|
| | | Intro | Inter | Comp | All | Intro | Inter | Comp | All | Intro | Inter | Comp | All |
| Codex | 12B | 4.14 | 0.14 | 0.02 | 0.92 | 9.65 | 0.51 | O.09 | 2.25 | 25.02 | 3.70 | 3.23 | 7.87 |
| AlphaCode | 1B | – | – | – | – | – | – | – | – | 17.67 | 5.24 | 7.06 | 8.09 |
| GPT-3 | 175B | 0.20 | 0.03 | 0.00 | 0.06 | – | – | – | – | – | – | – | – |
| GPT-2 | 0.1B | 1.00 | 0.33 | 0.00 | 0.40 | 2.70 | 0.73 | 0.00 | 1.02 | – | – | – | – |
| GPT-2 | 1.5B | 1.30 | 0.70 | 0.00 | 0.68 | 3.60 | 1.03 | 0.00 | 1.34 | 25.00 | 9.27 | 8.80 | 12.32 |
| GPT-Neo | 2.7B | 3.90 | 0.57 | 0.00 | 1.12 | 5.50 | 0.80 | 0.00 | 1.58 | 27.90 | 9.83 | 11.40 | 13.76 |
| CodeT5 | 60M | 1.40 | 0.67 | 0.00 | 0.68 | 2.60 | 0.87 | 0.10 | 1.06 | – | – | – | – |
| CodeT5 | 220M | 2.50 | 0.73 | 0.00 | 0.94 | 3.30 | 1.10 | 0.10 | 1.34 | – | – | – | – |
| CodeT5 | 770M | 3.60 | 0.90 | 0.20 | 1.30 | 4.30 | 1.37 | 0.20 | 1.72 | – | – | – | – |
| CodeRL+CodeT5 | 770M | 4.90 | **1.06** | **0.5** | 1.71 | 8.60 | **2.64** | 1.0 | 3.51 | **36.10** | 12.65 | 13.48 | 17.50 |
| PPOCoder +CodeT5 | 770M | **5.20** | 1.00 | **0.5** | **1.74** | **9.10** | 2.50 | **1.20** | **3.56** | 35.20 | **13.35** | **13.90** | **17.77** |

encoder-decoder architecture of CodeT5 outperforms larger models, and PPOCoder with CodeT5 further improves performance, surpassing even larger pre-trained LMs such as GPTs. As demonstrated in Table 3, PPOCoder +CodeT5 exhibits comparable or even superior *pass@k* performance than CodeRL+CodeT5, another RL-based fine-tuning mechanism specifically designed for the program synthesis task.

To further evaluate the generalizability of these models, the zero-shot performance of the APPS fine-tuned models was examined on the MBPP (Austin et al., 2021) program synthesis benchmark, which is a collection of 974 short (one sentence) problems, each including 1 correct Python solution and 3 corresponding unit tests. Table 4 shows the results of program synthesis on the MBPP benchmark. Both RL-based methods, PPOCoder +CodeT5 and CodeRL+CodeT5, fine-tuned on APPS, exhibit remarkable zero-shot performance on MBPP with a *pass@80* of 63% and 68.2%, respectively, surpassing even the largest GPT-137B's performance of 61.4%. As observed in Table 4, the proposed PPOCoder +CodeT5 outperforms CodeRL+CodeT5 on MBPP by a significant margin of 5.2%. This can be attributed to two factors. Firstly, CodeRL integrates the supervised cross-entropy loss with the RL policy gradient objective during the fine-tuning phase on APPS to prevent deviation from the pre-trained priors and avoid catastrophic forgetting. However, overoptimizing the token-matching cross-entropy during fine-tuning increases the chance of memorization on the APPS data and leads to inferior performance on unseen data like MBPP. PPOCoder regulates deviation by employing the KL-divergence penalty for generation instead. We can observe that this decreases the likelihood of memorization, resulting in improved generalizability of APPS fine-tuned model on the unseen MBPP benchmark. Secondly, CodeRL utilizes the actor-critic algorithm with REINFORCE reward policy gradient objective, while PPOCoder employs the PPO algorithm with an actor-critic *advantage* policy gradient objective and a trust region mechanism to ensure minimal deviations from the previous policy. This leads to a more stable and generalizable model optimization for new environments. One of the main motivations of PPOCoder is its potential to enhance the competitiveness of smaller models. As demonstrated, even with a model size of 770M, PPOCoder +CodeT5 surpasses the *pass*@80 scores achieved by larger models like GPT3-137B. This highlights the value of leveraging execution and structure alignment signals, which is considerably cheaper than training a model with a significantly larger size. Further comparisons and discussions with SOTA larger models, such as PaLM (Chowdhery et al., 2022) and CodeGen (Nijkamp et al., 2023) at various sizes, are provided in Appendix E.

## 4.4 Ablation Study

To investigate the effect of different components of PPOCoder, we conduct ablation experiments with several variants of our model, including different reward terms, RL objective/loss terms, divergence penalty, action space size, and the number of synthetic samples. We take the Java-Python translation as a case study and present the results in Fig. 3. More ablation studies are provided in Appendix D.

**Reward Elements.** Fig. 3(a) shows the effect of including different reward guidance components in the performance of PPOCoder. Models tested include CodeT5 without RL training, and with RL training utilizing different combinations of reward terms: *cs* (compiler feedback), ***dfg*** (semantic matching score from DFGs), and ***ast*** (syntactic matching score from ASTs), assuming the inclusion of divergence penalty by default in all the variants. Fig. 3(c) also represents the performance of model with compiler signal feedback (*+cs*) with and without the KL-divergence penalty. Results of Fig. 3(c) demonstrate that upon including

Table 4: Results of the zero-shot transferability on MBPP. Both zero-shot models are finetuned on APPS and evaluated on MBPP in the zero-shot setting.

| Model | Size | State | ↑*pass@80* |
|---|---|---|---|
| GPT | 224M | fine-tuned | 7.2 |
| GPT | 422M | fine-tuned | 12.6 |
| GPT | 1B | fine-tuned | 22.4 |
| GPT | 4B | fine-tuned | 33.0 |
| GPT | 8B | fine-tuned | 40.6 |
| GPT | 68B | fine-tuned | 53.6 |
| GPT | 137B | fine-tuned | 61.4 |
| CodeT5 | 60M | fine-tuned | 19.2 |
| CodeT5 | 220M | fine-tuned | 24.0 |
| CodeT5 | 770M | fine-tuned | 32.4 |
| CodeRL+CodeT5 | 770M | zero-shot | 63.0 |
| PPOCoder +CodeT5 | 770M | zero-shot | **68.2** |

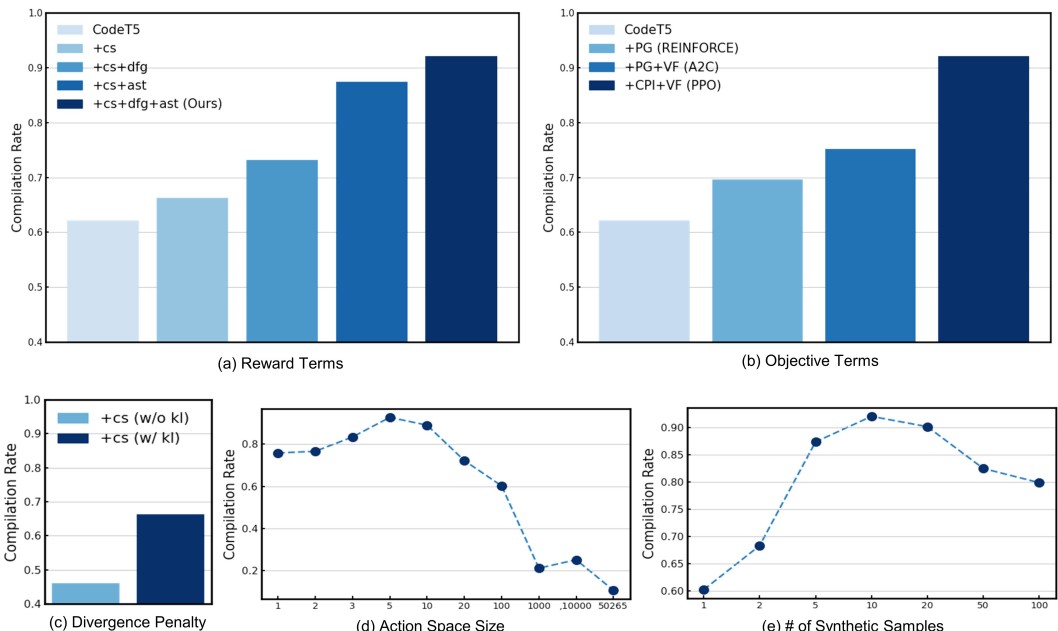

Figure 3: Ablation experiment results on Java-Python translation with different configurations of (a) reward, (b) loss, (c) divergence penalty, (d) action space size, and (e) number of synthetic samples.

the KL-divergence penalty, we see a marked improvement in the compilation rate performance, indicating the necessity of divergence constraint in the RL-based fine-tuning due to the broad exploration space and the high potential of policy deviation from the pre-trained priors (i.e., catastrophic forgetting). Fig. 3(a) also shows that compiler signal is beneficial in the improvement of model performance. However, the improvement obtained by $+cs$ is still minor, potentially due to the sparsity of discrete compiler feedback received at the end of the generation episode. Results show that integrating feedback with the syntactic/semantic matching score further boosts performance , with the syntactic matching score ($+cs+ast$) offering more improvement in the compilation rate than the semantic matching score ($+cs+dfg$). Finally, the best performance is achieved by utilizing all four reward elements, indicating the importance of incorporating multiple sources of feedback in the learning process.

**Loss Elements.** Fig. 3(b) represents the results of PPOCoder with different objective configurations. We observe that the policy gradient objective alone ($+PG$), i.e., the REINFORCE algorithm, can boost the performance of the CodeT5 model. The compilation rate further improves by introducing the value function as critic ($+PG+VF$), i.e., A2C algorithm. Results show that the best performance is achieved by utilizing proximal conservative policy iteration with value optimization ($+CPI+VF$), indicating that the Proximal Policy Optimization (PPO) algorithm performs superior to others in this framework.

**Action Space Size.** We examine the effectiveness of action space size on PPOCoder's performance by adjusting the $k$ parameter in the *top-k* policy synthetic sampling. Fig. 3(d) shows that when $k = 1$, PPOCoder may not be able to have enough exploration for better possible policy updates. On the other hand, when $k$ gets too large, PPOCoder may become overwhelmed by many different possible actions and struggle to learn the optimal policy, leading to degraded performance. Therefore, results reveal that a small value of $k$ ($k = 1$) may not provide sufficient exploration, while a large value ($k = 50265$ (vocab size)) can hinder the learning of optimal policy. In our experiments, we usually use the action space size 5 which provides a good balance for optimal exploration in most cases.

**No. of Synthetic Samples.** The effect of synthetic policy sample size on PPOCoder's performance is examined by modifying the *num_samples* in Alg. 1. Fig. 3(e) shows that an increase in *num_samples* from 1 to 10 improves performance, but further increases lead to a slight decline in performance. This indicates that while more synthetic samples might improve pattern detection, excessive synthetic samples can reach saturation, becoming less effective and adding computational complexity.

## 5 Conclusion

We developed a PPO-based deep reinforcement learning framework for improving the quality of code generated by PL models. We identified some limitations of traditional objective functions for code generation tasks and designed a new optimization framework that is geared towards PLs as opposed to natural language. We incorporated compiler feedback and unit tests along with syntactic and semantic related qualitative feedback into our RL framework to encourage the model to generate more syntactically and logically correct codes. Results of our experiments show the effectiveness of our method compared to baselines in improving the syntactic/functional correctness of the generated codes. The ablation experiments also show the impact of various model components on the final performance. It is important to note that one of the limitations of PPOCoder is the added computational time required for RL-based optimization and the possibility that it may not yield significant improvements on other evaluation metrics not directly targeted during RL optimization. However, PPOCoder is primarily motivated by its ability to enhance the performance of smaller models which is a more cost-effective strategy than training significantly larger ones.

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

## Appendix

## A  Implementation Details

In all our experiments, we employ batch size of 32, AdamW optimizer with a weight decay of 0.05, and a learning rate that warms up from $1e-7$ to $2e-5$ over the first 1000 steps, then decays based on the inverse square root of the number of steps, as outlined in (Loshchilov & Hutter, 2019). We use the `tree-sitter` parsing library [4] to identify AST sub-trees for code written in different languages. To construct the data-flow graph (DFG), we first identify variables by using the leaves of the AST, and then create directed edges

---

[4] `https://github.com/tree-sitter/tree-sitter`

between the variables based on their relations. For the syntactic (semantic) matching, we follow a very simple mechanism adopted in (Ren et al., 2020). After AST/DFG creation for both the generated code candidate and the reference target code, we examine each sub-tree/data-flow with the corresponding depth in the reference AST/DFG. If a match is found in the candidate, we increment the matching sub-trees/data-flows count and eliminate the current sub-tree/data-flow from the list of all reference ones. All of our experiments are implemented with PyTorch and trained using 4 Quadro RTX 8000 GPUs, with 48GB of RAM.

## A.1 Code Completion

As explained before, we conduct code completion experiments on the CSN [5] dataset with Python programs. For Python compilation, we adopt `py_compile` [6] library. In the experiment result table, The BiLSTM baseline is the Seq2Seq Bi-directional LSTM model taken with the default settings used in (Luong et al., 2015). The transformer baseline is a 6-layer transformer decoder as used in (Vaswani et al., 2017). BiLSTM and transformer baselines are not pre-trained and will be trained from the random initialization on this task. GPT-2 baseline (Radford et al., 2019) is a decoder-only transformer and is pre-trained on a large-scale text corpus. CodeGPT (Lu et al., 2021) and CodeT5 (Wang et al., 2021) baselines are both pre-trained on a large-scale code corpus with the default GPT-2 (decoder-only) and T5 (encoder-decoder) architectures. The pre-trained CodeGPT and CodeT5 models are also fine-tuned on this task, and the PPOCoder +CodeT5 is initialized from the fine-tuned CodeT5. The fine-tuning of the CodeT5 model with token-level matching objectives is stopped when convergence is reached, typically after around 100 epochs. PPOCoder is implemented with the discount rate $\gamma = 1$, KL divergence penalty coefficient $\beta = 0.1$, policy ratio clip range $\epsilon = 0.2$ and the value error coefficient $\alpha = 0.001$. To sample synthetic hypothesis from the stochastic policy, we use the *top-k* sampling with $k = 5$ as the action space size. We are training PPOCoder +CodeT5 with *num_samples* $= 3$ as the number of synthetic samples generated for each sample of the CSN dataset. Therefore, PPOCoder observes 40K×3=120K input-output sample pairs with synthetic outputs during RL optimization for this task. In all code completion experiments on CSN, we set the maximum source and target sequence length as 400, and the maximum number of RL optimization epochs as 6.

## A.2 Code Translation

We conduct code translation experiments on the XLCoST dataset with programs in six parallel PLs (C++, Java, Python, C#, PHP, and C). For testing Python compilation, we use the `py_compile` module that comes built-in with Python version 3.8.0. For Java compilation, we use the `javac` compiler, version 1.8.0. We use `gcc` version 7.5.0 for C and C++ compilations. Syntax checking for PHP is performed using the `php -l` command, PHP version 7.2.24. C# compilation is also checked using the Mono C# compiler, version 4.6.2.0. In the result table of code translation experiments on XLCoST, CodeBERT baseline is an encoder-only pre-trained transformer model; PLBART and CodeT5 are encoder-decoder pre-trained transformer models. All these models (i.e., CodeBERT, PLBART, and CodeT5) are first pre-trained on a large-scale code corpus and then fine-tuned on XLCoST for the code translation task of each language pair. Also, in PPOCoder +CodeT5, the policy network in each language-pair translation is initialized from a CodeT5 model fine-tuned on that language-pair. Here, in the implementation of PPOCoder, we use discount rate $\gamma = 1$, KL divergence penalty coefficient $\beta = 0.01$, policy ratio clip range $\epsilon = 0.2$ and the value error coefficient $\alpha = 0.01$. To sample from the stochastic policy, we use the *top-k* sampling with $k = 5$ as the action space size. Also, we train PPOCoder +CodeT5 with *num_samples* as 100 for low-resource experiments with $C$ as the source/target language; 20 for experiments with PHP as source/target language; and 10 for other translation experiments. For example, for each source code $x$ in Java-Python translation, we generate 10 synthetic samples based on the policy, thus, $4,811 \times 10 = 48,110$ (based on Table 6) input-output pairs are used in total to train PPOCoder for the Java-Python translation. We also set the maximum number of epochs for RL optimization as 15, and the maximum source and target sequence lengths to 400 for all code translation tasks on XLCoST.

## A.3 Program Synthesis

We conduct program synthesis experiments on the APPS dataset with the pair of NL problem descriptions and Python programs in three difficulty levels: Introductory, Interview, and Competition. In the results of

---

[5] `https://github.com/github/CodeSearchNet#data-details`

[6] `https://docs.python.org/id/3.6/library/py_compile.html`

program synthesis experiments on APPS, GPT-3, and Codex baselines are tested on the APPS dataset in the few-shot settings without fine-tuning, while other baselines are fine-tuned on the APPS with the cross-entropy loss. The PPOCoder +CodeT5 is also initialized from the fine-tuned CodeT5 on the APPS program synthesis. We use the *pass@k* metric to evaluate the functional correctness of a program, where a code is considered correct if it successfully passes all unit tests designed for the specific problem. In the PPOCoder's implementation on APPS, we use discount rate $\gamma = 1$, KL divergence penalty coefficient $\beta = 0.05$, policy ratio clip range $\epsilon = 0.2$, and the value error coefficient $\alpha = 0.001$. To sample synthetic hypothesis from the stochastic policy, we use the *top-k* sampling with $k = 5$ as the action space size. We are training PPOCoder +CodeT5 with *num_samples* $= 5$ as the number of synthetic samples used for each APPS problem. In all the program synthesis experiments on APPS, we set the maximum source and target sequence lengths as 600 and 512, and the maximum number of RL optimization epochs as 8. We have also evaluated performance of PPOCoder +CodeT5 on the MBPP dataset in a zero-shot setting. We compared the zero-shot performance with GPT and CodeT5 models fine-tuned on MBPP for 60 epochs with maximum source and target sequence lengths of 400.

## B  Additional XLCoST Dataset Details

We are using the XLCoST [7] (Zhu et al., 2022a) dataset for code translation experiments with different language pairs among six parallel PLs (C++, Java, Python, C#, PHP, and C). XLCoST has been created by scraping solutions off the popular programming tutorial and interview preparation website GeeksforGeeks[8]. This kind of multilingual parallel data is perfect for training translation models. The dataset statistics are provided in Table 5. This dataset is parallel at both program snippet and full program levels. The upper triangle of Table 5 summarizes snippet-level statistics, and the lower triangle summarizes the program-level statistics. For the purpose of our experiments in the evaluation of syntactic/functional correctness, we only use program-level data. It is noteworthy that all programs in this dataset do not successfully compile. Therefore, in our experiments, we only use the compilable filtered parallel data in both source and target language pairs. Our hypothesis is that using ground truth data that is free of compilation errors will provide a stronger signal to PPOCoder for correcting syntactic errors than using the full data, which could also contain flawed programs. To do so, we designed an automated compilation pipeline that evaluates which programs in the dataset compile without any errors. Table 6 shows statistics of the compilable filtered dataset for all languages (except JavaScript which is excluded due to the lack of compilation).

Table 5: XLCoST dataset statistics. The upper triangle (in bold font) shows the number of parallel code snippets, and the lower triangle shows the number of parallel programs. **JS** is short for Javascript.

| Lang | | C++ | Java | Py | C# | JS | PHP | C |
|---|---|---|---|---|---|---|---|---|
| **C++** | train | – | **89040** | **80100** | **85662** | **69507** | **17811** | **3386** |
| | val | – | **4419** | **3913** | **4408** | **3808** | **923** | **352** |
| | test | – | **8059** | **7228** | **7922** | **6965** | **1647** | **222** |
| **Java** | train | 9450 | – | **77759** | **87065** | **69341** | **17853** | **2996** |
| | val | 490 | – | **3938** | **4437** | **3826** | **929** | **353** |
| | test | 901 | – | **7259** | **8011** | **7005** | **1672** | **238** |
| **Py** | train | 9139 | 8991 | – | **75843** | **67219** | **17616** | **2478** |
| | val | 468 | 471 | – | **3922** | **3750** | **923** | **311** |
| | test | 878 | 882 | – | **7215** | **6861** | **1655** | **203** |
| **C#** | train | 9187 | 9301 | 8826 | – | **68093** | **17873** | **2958** |
| | val | 488 | 491 | 470 | – | **3826** | **928** | **352** |
| | test | 890 | 898 | 877 | – | **6961** | **1668** | **238** |
| **JS** | train | 8482 | 8470 | 8182 | 8367 | – | **17117** | **1875** |
| | val | 472 | 475 | 459 | 475 | – | **921** | **309** |
| | test | 878 | 881 | 864 | 877 | – | **1617** | **200** |
| **PHP** | train | 3056 | 3068 | 3003 | 3071 | 2971 | – | **856** |
| | val | 157 | 158 | 153 | 158 | 157 | – | **271** |
| | test | 303 | 307 | 304 | 307 | 302 | – | **183** |
| **C** | train | 402 | 409 | 380 | 394 | 308 | 170 | – |
| | val | 59 | 59 | 59 | 59 | 59 | 55 | – |
| | test | 45 | 49 | 48 | 49 | 49 | 43 | – |

---

[7] https://github.com/reddy-lab-code-research/XLCoST
[8] https://www.geeksforgeeks.org

Table 6: XLCoST dataset statistics after filtering for compilable programs. Javascript is excluded.

| Lang | | C++ | Java | Python | C# | PHP | C |
|---|---|---|---|---|---|---|---|
| **C++** | train | – | – | – | – | – | – |
| | val | – | – | – | – | – | – |
| | test | – | – | – | – | – | – |
| **Java** | train | 5251 | – | – | – | – | – |
| | val | 266 | – | – | – | – | – |
| | test | 520 | – | – | – | – | – |
| **Python** | train | 5325 | 4811 | – | – | – | – |
| | val | 266 | 250 | – | – | – | – |
| | test | 529 | 496 | – | – | – | – |
| **C#** | train | 5464 | 5081 | 5027 | – | – | – |
| | val | 279 | 262 | 259 | – | – | – |
| | test | 553 | 513 | 529 | – | – | – |
| **PHP** | train | 1758 | 1595 | 1785 | 1764 | – | – |
| | val | 91 | 85 | 93 | 92 | – | – |
| | test | 192 | 176 | 201 | 197 | – | – |
| **C** | train | 233 | 191 | 169 | 204 | 92 | – |
| | val | 34 | 34 | 32 | 34 | 33 | – |
| | test | 28 | 28 | 24 | 29 | 26 | – |

# C   Additional Evaluation of Code Translation on TransCoder Benchmark

For the code translation task, we additionally evaluate the performance of PPOCoder using the TransCoder[9] benchmark (Roziere et al., 2020). This benchmark comprises 852 parallel functions in C++, Java, and Python PLs, along with some unit tests for certain functions to assess the operational accuracy of the generated translations. There are a few key points to consider in these experiments: (*i*) The TransCoder benchmark has 466, 481, and 463 test functions with unit tests for C++, Java, and Python, respectively. This number of samples is inadequate for reinforcement learning (RL)-based model fine-tuning. As depicted in Table 6 of Appendix B, XLCoST contains over 5000 training samples for each of these three high-resource languages, which is ample for RL-based fine-tuning. Therefore, we employ the TransCoder data only as an evaluation benchmark for models trained on XLCoST data for each language pair. (*ii*) We also observed that XLCoST and TransCoder both stem from the GeeksforGeeks (GfG) website, resulting in some overlap. Consequently, we will assess the performance only on the filtered TransCoder test functions that do not coincide with the XLCoST training data. Upon examination, we discovered that approximately 30% of TransCoder test functions overlap with XLCoST. The detailed statistics for the different PLs are presented in Table 7.

Table 7:  Statistics of overlapping TransCoder functions with XLCoST.

| | C++ | Java | Python |
|---|---|---|---|
| # of functions with unit tests | 466 | 481 | 463 |
| # of overlapping functions with XLCoST | 137 | 169 | 139 |
| % of overlap | 29.39 | 35.13 | 30.02 |

Table 8 displays the evaluation results on the TransCoder benchmark, including metrics such as computational accuracy, compilation error, runtime error, failed unit tests, and timeouts. The evaluation experiments on the TransCoder benchmark reveal that PPOCoder+CodeT5 enhances CodeT5's computational accuracy across various programming languages: 39.9 to 72.1, 36.8 to 57.1, 45.2 to 65.6, 37.4 to 52.1, 26.1 to 39.0, and 15.9 to 51.1 for C++→Java, C++→Python, Java→C++, Java→Python, Python→C++, and Python→Java translations, respectively. Upon examining the failed examples, we found that PPOCoder reduces compilation errors from 42.2 to 17.4, 25.2 to 1.1, 47.9 to 19.7, 35.3 to 0.2, 59.9 to 33.5, and 48.3 to 7.8, demonstrating the efficacy of involving compiler signal in the model fine-tuning. The PPOCoder+CodeT5 results also highlight several intriguing observations: (1) Many failures are due to compilation errors, particularly when the target language is C++, followed by Java. (2) PPOCoder achieves greater compilability improvements for Python and Java target PLs as compared to C++. This trend was also observed in the XLCoST dataset translations. As previously mentioned, this is likely attributable to the lower-level constructs in C++, which could introduce more error sources. (3) Runtime errors predominantly occur when Python is the target PL, which can be attributed to Python's more permissive compilation process compared to other languages. Most of the remaining errors arise from the program producing incorrect output (i.e., failing at least one

---

[9]https://github.com/facebookresearch/TransCoder

unit test). Timeout errors are generally caused by infinite loops, which appear to be more prevalent in translations between Java and Python PLs.

Table 8: Results of code translation evaluation on the TransCoder benchmark. Both models are fine-tuned on XLCoST and evaluated on TransCoder functions with unit tests. The "Overall" column shows the weighted average scores over all language pairs.

| Model | Metric | C++→Java | C++→Python | Java→C++ | Java→Python | Python→C++ | Python→Java | Overall |
|---|---|---|---|---|---|---|---|---|
| **CodeT5** | %Computational Accuracy | 39.9 | 36.8 | 45.2 | 37.4 | 26.1 | 15.9 | 31.9 |
| | %Compile Error | 42.4 | 25.2 | 47.9 | 35.5 | 59.9 | 48.3 | 43.2 |
| | %RunTime Error | 6.8 | 25.1 | 1.8 | 21.4 | 1.4 | 9.0 | 11.1 |
| | %Failed a Unit Test | 7.0 | 11.2 | 4.2 | 6.8 | 11.1 | 19.4 | 9.95 |
| | %Timeout | 3.9 | 1.7 | 0.9 | 8.9 | 1.5 | 7.4 | 4.05 |
| **PPOCoder + CodeT5** | %Computational Accuracy | 72.1 | 57.1 | 65.6 | 52.1 | 39.0 | 51.1 | 56.3 |
| | %Compile Error | 17.4 | 1.1 | 19.7 | 0.2 | 33.5 | 7.8 | 13.3 |
| | %RunTime Error | 3.2 | 28.1 | 2.8 | 20.1 | 3.5 | 11.2 | 11.5 |
| | %Failed Unit Test | 5.9 | 13.0 | 11.4 | 10.9 | 23.2 | 15.9 | 13.4 |
| | %Timeout | 1.4 | 0.7 | 0.5 | 16.7 | 0.8 | 13.6 | 5.6 |

## D  Additional Ablation Experimetns

Similar to the ablation results presented in the Fig. 3 for the Java-Python code translation, Fig. 4 shows the results of additional ablation experiments on the Python-C# translation task for different reward terms, RL objective terms, action space size, and the number of synthetic samples.

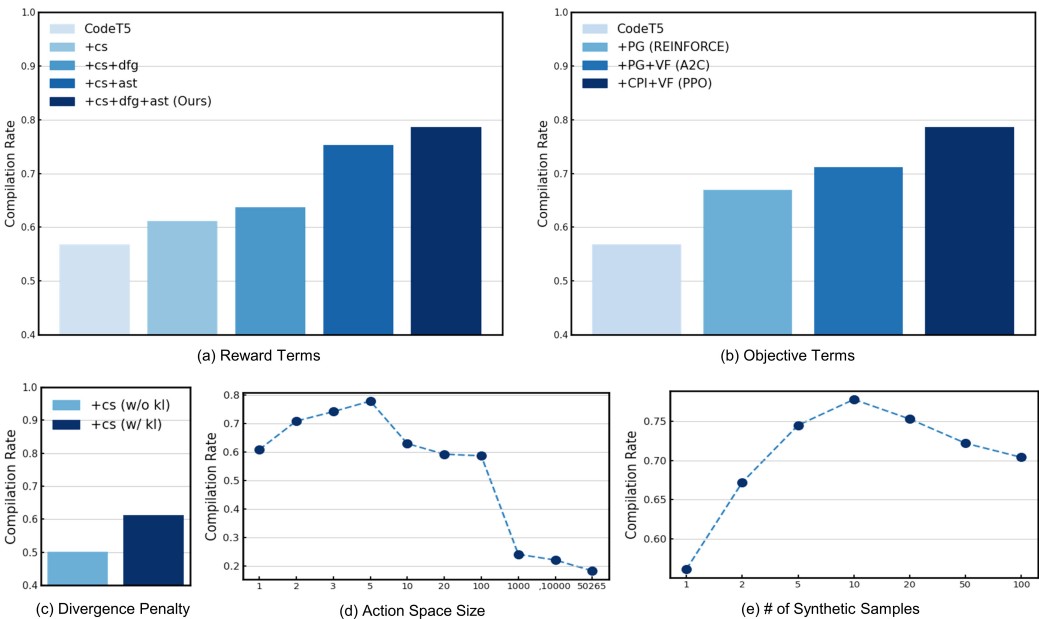

Figure 4: Ablation experiment results on Python-C# translation with different configurations of (a) reward, and (b) loss, (c) divergence penalty, (d) action space size, and (e) number of synthetic samples.

Fig. 5 shows the additional ablation experiment conducted (on the same Java-Python translation case study) for the model variants without the compiler signal (**+cs**) in reward. Results clearly show that model variants with the compiler signal (**+cs+dfg**: 0.732, **+cs+ast**: 0.874, **+cs+dfg+ast**: 0.921) achieve better compilation rates compared to model variants without the compiler signal (**+dfg**: 0.674, **+ast**: 0.801, **+dfg+ast**: 0.879), with the default inclusion of the KL-divergence penalty in all models. Therefore, the sparse compiler signal feedback received at the end of the generation episode is not detrimental but, in fact, beneficial when integrated into the reward function for model optimization.

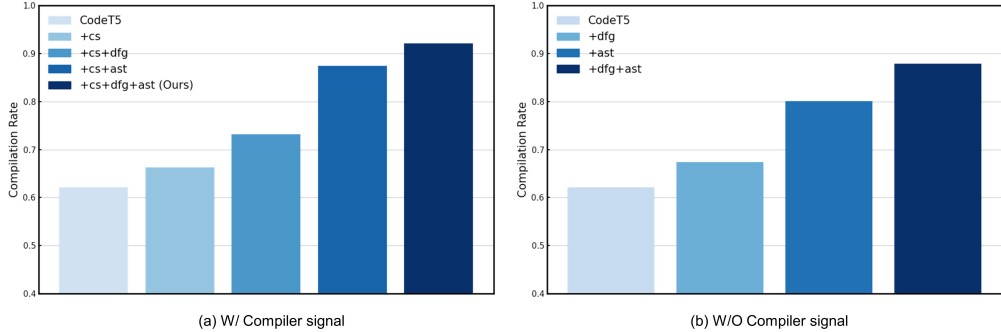

Figure 5: Ablation experiment results on Java-Python translation for the model variants (a) with compiler signal (w/ **+cs**), and (b) without compiler signal (w/o **+cs**).

Figure 6 presents the ablation experiment results for the program synthesis task on the MBPP dataset for our model with different (a) reward components and (b) RL objective terms. All model variations shown here are fine-tuned on APPS (Hendrycks et al., 2021) and evaluated in the zero-shot setting on MBPP. Figure 6(a) indicates that using only the compiler signal (**+cs**) increases $pass@80$ by approximately 16%. Additionally, incorporating syntactic and semantic matching scores into the reward function can further enhance performance. The results of model variants with different reward components in Fig. 6 are reported with the inclusion of the KL-divergence penalty as default. Figure 6(b) also demonstrates that all RL algorithms can effectively improve the base CodeT5 model's performance. The best results are achieved by employing proximal conservative policy iteration with value optimization (**+CPI+VF**), indicating the outperformance of using PPO compared to other RL algorithms in this framework.

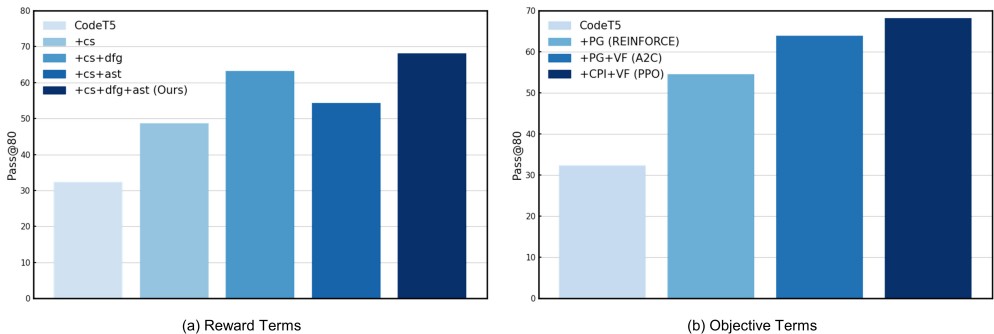

Figure 6: Ablation experiment results on MBPP program synthesis with different configurations of (a) reward, and (b) loss. All model variants are finetuned on APPS and evaluated on MBPP in the zero-shot setting.

Fig. 7 illustrates the relationship between the compilation rate and $pass@1$ performance on MBPP program synthesis across various model variants with different configurations of reward terms. As it can be observed, there is a positive correlation between these two metrics, indicating that an increased model performance in terms of pass@1 can be associated with better model performance in terms of compilation rate. Specifically, the base CodeT5 model, with a compilation rate of 50.1%, achieved a $pass@1$ rate of 3.4. When we added compiler signal (**+cs**), the compilation rate improved significantly to 71.2%, leading to an impressive increase in $pass@1$ rate to 16.4. A similar positive trend was observed when further reward components were incorporated. The model variant **+cs+dfg**, which uses data-flow graph, registered a higher compila-

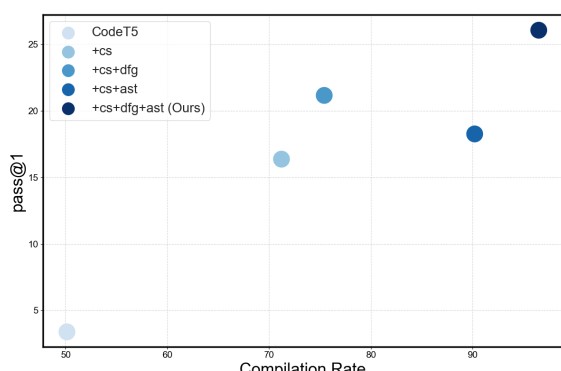

Figure 7: Computation accuracy ($pass@1$) vs. compilation rate performance between PPOCoder +CodeT5 with different configurations of reward on MBPP program synthesis.

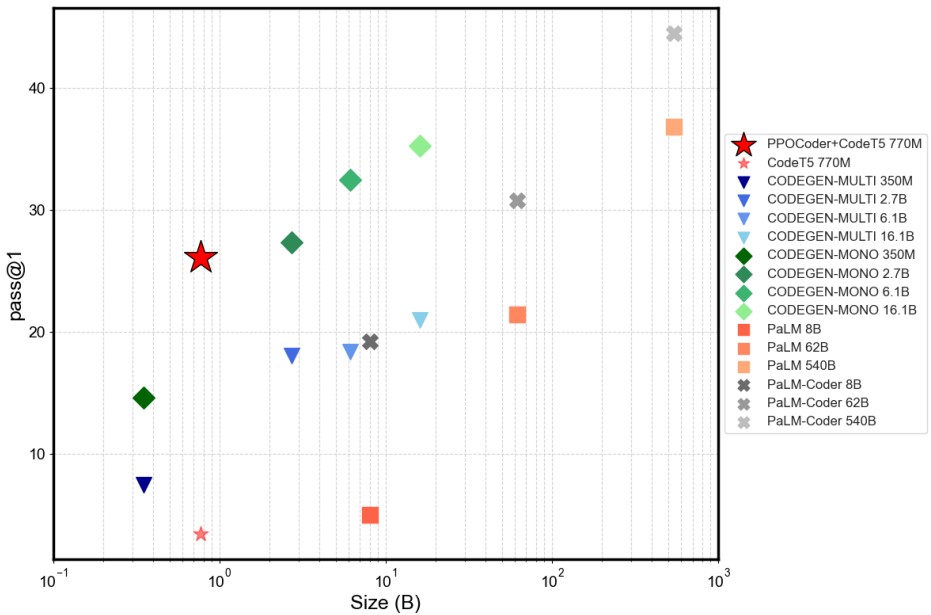

Figure 8: Pareto comparison of performance ($pass$@1) vs. model size on MBPP between PPOCoder +CodeT5 and larger SOTA models such as CodeGen and PaLM with various size, highlighting the performance-resource trade-off.

tion rate of 75.4%, and correspondingly, a better $pass$@1 rate of 21.2. Even though the addition of abstract syntax trees (**+cs+ast**) yielded a higher compilation rate of 90.2%, the $pass$@1 rate was slightly lower at 18.3, likely due to other complex factors influencing model performance. Our final proposed model variant, **+cs+dfg+ast**, yielded the best results with the highest compilation rate of 96.5% and the most elevated $pass$@1 rate of 26.1. Fig. 7 demonstrates that there's generally a positive correlation between higher compilation rates and better pass@1 performance.

## E  Discussion and Comparison with SOTA Large Models

Fig. 8 presents a comprehensive comparison between PPOCoder +CodeT5 and prominent large language models such as PaLM (Chowdhery et al., 2022) and CodeGen (Nijkamp et al., 2023) for the program synthesis task on the MBPP (Austin et al., 2021) dataset. The vertical axis represents the $pass$@1, which indicates the percentage of problems for which all unit tests passed after sampling only one program from the model ($k = 1$). The Pareto chart showcases the performance-resource trade-off by comparing the model size and the corresponding $pass$@1 scores. We observe the presence of pre-trained PaLM models, denoted by the PaLM legend, as well as fine-tuned PaLM-Coder models. As expected, the fine-tuned PaLM-Coder models exhibit superior performance in MBPP program synthesis compared to the pre-trained PaLM models. PaLM models are available in three different sizes: 8B, 62B, and 540B parameters, with performance improving as the model size increases. Notably, the largest 540B model consistently performs the best. Additionally, the chart displays four different sizes of CodeGen models: 350M, 2.7B, 6.1B, and 16.1B parameters. The legend differentiates between multi-lingual pre-trained CodeGen models (CodeGen-Multi) and Python mono-lingual pre-trained models (CodeGen-Mono). The CodeGen-Mono models outperform the CodeGen-Multi models in MBPP program synthesis (on Python target PL).

Comparing the performance of PPOCoder +CodeT5 (26.1) with other models, we find that it surpasses the $pass$@1 scores of all CodeGen-Multi models (7.4, 18.0, 18.3, and 20.9) across varying sizes. Additionally, PPOCoder +CodeT5 outperforms PaLM-8B (5.0), PaLM-62B (21.4), and CodeGen-Mono-350M (14.5) in $pass$@1. However, it falls short of the performance achieved by the largest PaLM model with 540B parameters (36.8) and the larger CodeGen-Mono models with sizes of 2.7B, 6.1B, and 16.1B, which exhibit $pass$@1

scores of 27.3, 32.4, and 35.2, respectively. Analyzing the performance-size trade-off highlighted in Fig. 8, it becomes evident that PPOCoder +CodeT5 occupies the top-left position, denoting superior performance (higher *pass*@1) with the smallest possible model size. This placement on the first Pareto front aligns with one of the main motivations of PPOCoder, which is to enhance the competitiveness of smaller models. By utilizing execution and structure alignment signals, PPOCoder achieves remarkable performance gains while avoiding the resource-intensive nature of training significantly larger models. The findings from Fig. 8 reinforce the effectiveness of PPOCoder +CodeT5 and substantiate the advantageous performance-resource trade-off it offers.

## F  Case Studies

Fig. 9 presents a case study of PHP to Python translation for both CodeT5 and PPOCoder +CodeT5 models. We can observe that CodeT5 is unable to translate the code without compilation errors, and the use of PPOCoder enhances CodeT5's capability of generating compilable code in the target language while preserving the code fluency and distributions learned by the pre-trained CodeT5. For this example, CodeT5 faces some problems: (1) "Error: Invalid Syntax": The "for" loop should start from next line ; (2) "Error: Local variable 'i' referenced before assignment". The for loop index is "j" but index "i" is called inside the loop; and (3) Semantic Relations: The equivalent of PHP_INT_MAX may be 1000 (or `sys.maxsize`) in Python but CodeT5 translates it to `int(n/2)` which is logically wrong.

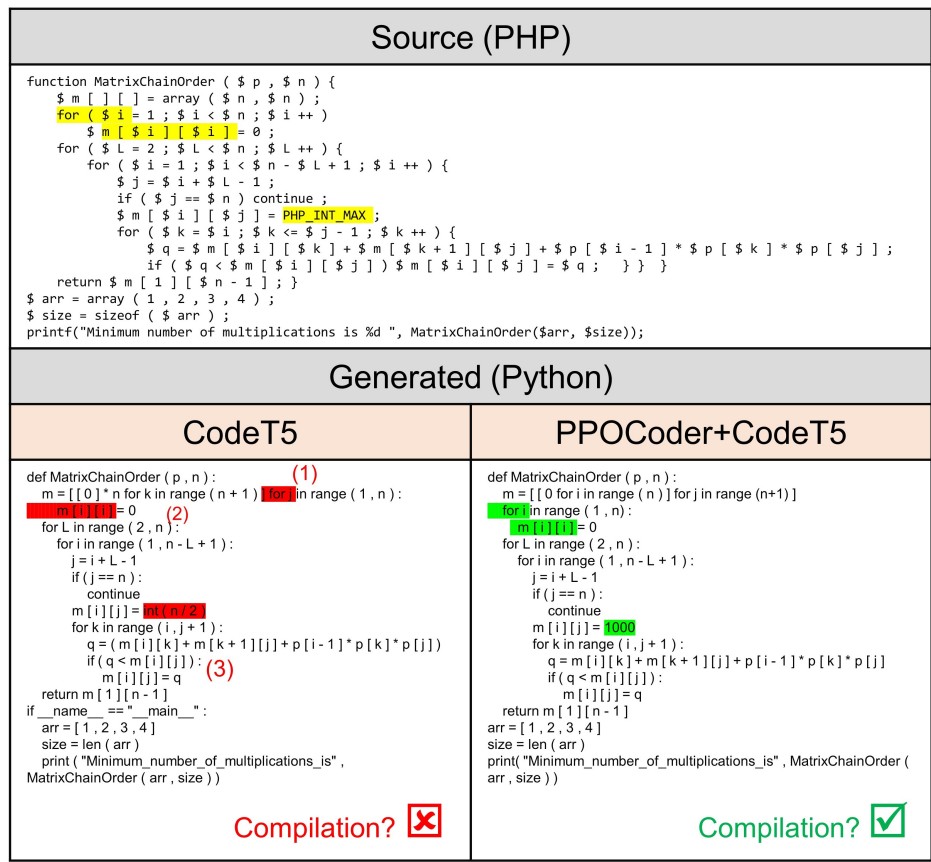

Figure 9: Case study example for PHP-Python code translation. The erroneous snippets are highlighted in red.

Fig. 10 shows an example of Java to C++ translation for both CodeT5 and PPOCoder +CodeT5. Similar to the previous case, it can be observed that the compilation is improved by PPOCoder. For this example, CodeT5's translation has these issues: (1) CodeT5 generates a non-standard data type called `subset` which

takes in a pair of integers. The use of the non-standard data-type without importing it or defining it causes a compilation error, while PPOCoder +CodeT5 generates the C++ translation using the correct `vector` data-type corresponding to `ArrayList` in the source Java; (2) "Error: Local variable 'i' referenced before assignment": for loop index is "j" but "i" is used later; and (3) "Error: Invalid Syntax" for missing "(" when calling the function.

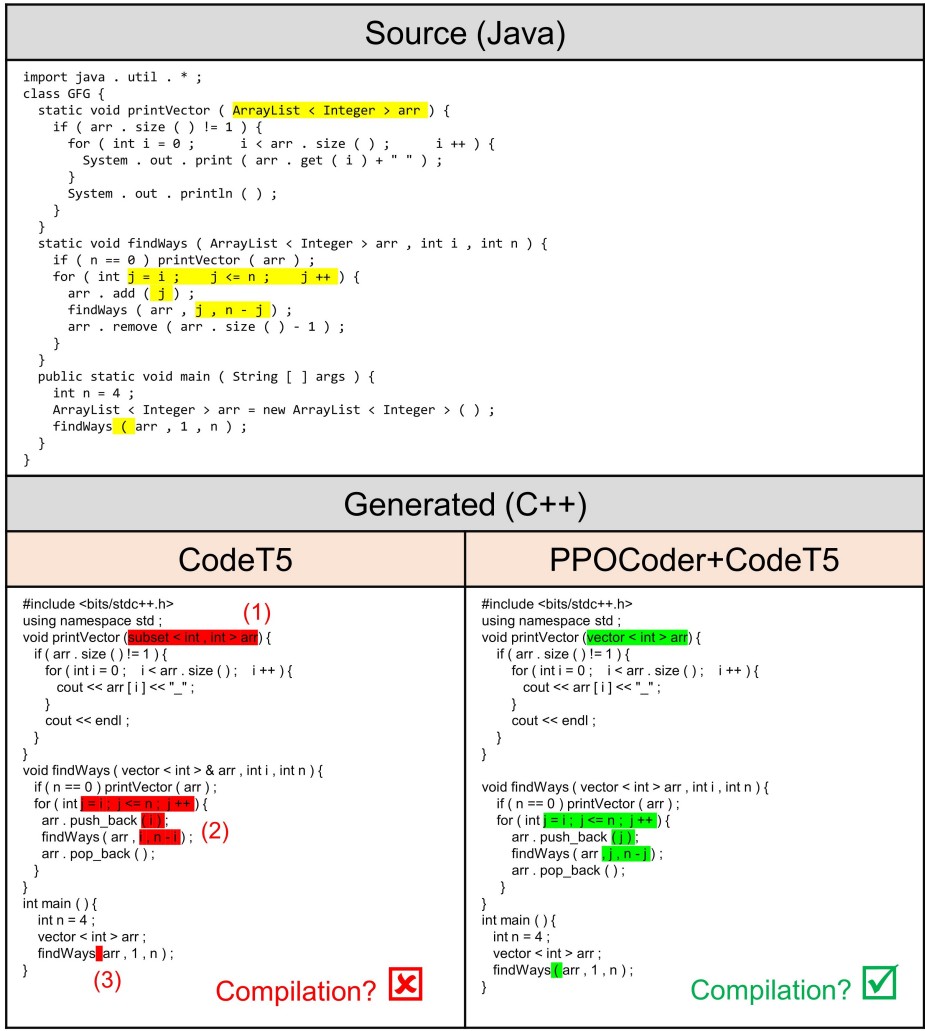

Figure 10: Case study example for Java-C++ code translation. The erroneous snippets are highlighted in red.

We also looked into an example of program synthesis from APPS. Fig. 11 illustrates a case study of Python program synthesis for a problem description given in NL. Although both CodeT5 and PPOCoder +CodeT5 generate compilable programs, CodeT5's generation cannot pass some hidden unit tests that capture different corners of the problem. These two generated codes are different in the highlighted places. As we can see, the CodeT5's generation (1) initializes the 2D array 'dp' with ones on both dimensions instead of zeros on one dimension; and (2) the for loop inside the nested loop uses the condition $k >= j$ instead of the correct condition $k <= j$. Although these differences do not produce any compilation errors, they result in logical errors and cause unit tests to fail.

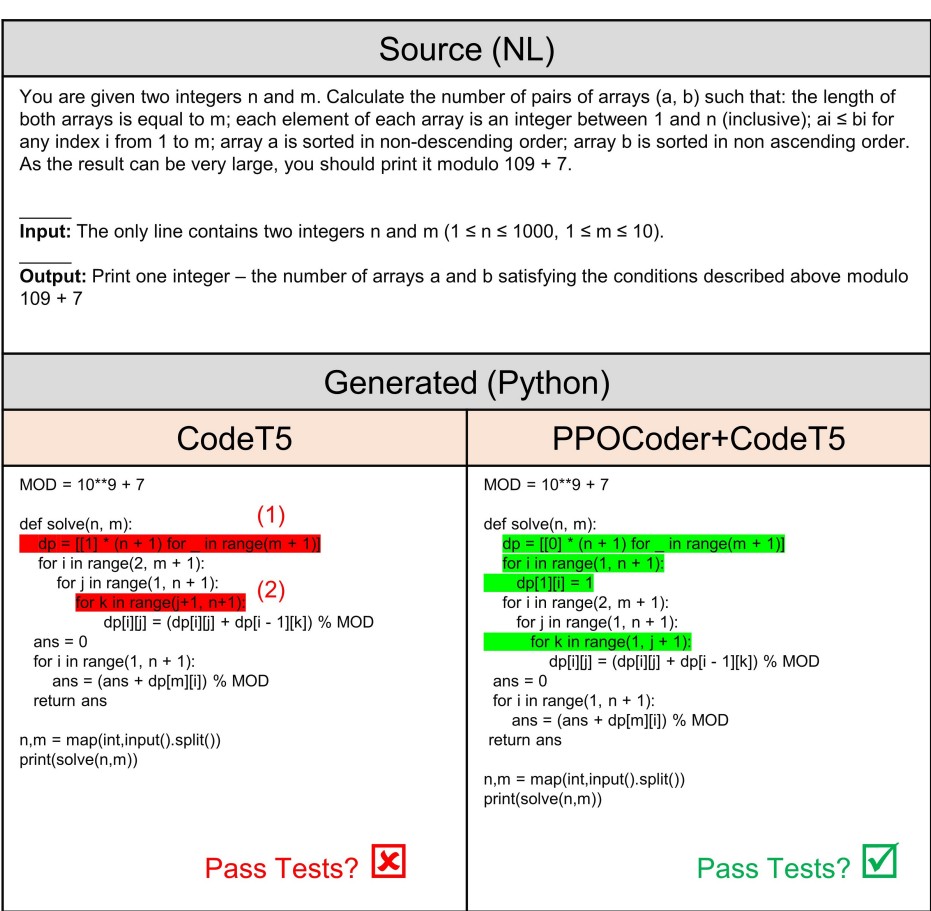

Figure 11: Case study example of APPS for Program Synthesis (NL2Code). The problematic snippets are highlighted in red.

