# OpenReview forum: "Execution-based Code Generation using Deep Reinforcement Learning"
_TMLR — Accepted by TMLR_

### Review · Reviewer_aDTf · 2023-05-09

**Summary Of Contributions:**

An adaptation of PPO to the code generation setting is presented. To this end, a reward function consisting of 4 terms (execution feedback; matching of AST subtrees; matching of data flow edges; divergence from underlying code model) is introduced. Experiments on three datasets (code completion, code translation, program synthesis) show that the method minimally improves on the underlying base model.

**Audience:**

Yes

**Claims And Evidence:**

No

**Requested Changes:**

- p2: "eight programming languages However," -> Missing "."
- Sect. 2.3 misattributes work (RLHF was not introduced by InstructGPT, as that paper itself says!) and contains sentences that are not parseable (what is "matching scores between the AST sub-trees and DFG edges" supposed to mean?). Please rework this.
- p6 (and following): $R_{dfg}$ has ugly spacing after the $f$. Please use $R_{\mathit{dfg}}$ (R_{\mathit{dfg}}).
- Eq (7): Please reformat so that the is-end-of-token term is on one line, and the other case on the next one - this is unnecessarily hard to parse.
- Sect. 3.2.1: Given that paper explicitly discusses Python and PHP: What does "compiler signal" mean here - that the code is successfully parsed by python/php? Is type checking used?
- Overall, Sect. 3 spends substantial space on recapping fairly standard RL/PPO bits, and misses many details on how the reward function (the core contribution of the paper!) is actually implemented. Please rearrange this, moving standard components into appendix and expanding on the novel contribution.
- Sect. 3.2.2: The choice of $R_{ast}$ is not explained - why is token similarity not sufficient? Using the AST explicitly violates the author's aim to be language-agnostic, after all. The given intuition ("token missing and data type errors") would apply on the token level as well. Please explain.
- Sect. 3.2.3: This section misses very important details, foremost how the DFG is computed (given that this is non-trivial in heap-using programs with aliasing) and how DFG edges are matched (finding a graph isomorphism is after all very much non-trivial). Please explain.
- Sect. 4.1: Details such as the size of the used CodeT5 model (60M? 220M? 770M?) are missing. Please add.
- p10: "absence of pointers and memory management constructs, which can be a source of errors in languages like C++ and C#." - C# has neither pointers nor explicit memory management, please fix this.
- Table 3: Explain what the "All" column represents.

**Strengths And Weaknesses:**

* (~) p1: "given that the generated code is intended for machine execution as opposed to human comprehension": I fundamentally disagree with this statement. Any code meant to be used more than once is also intented to be read by other humans, which has substantial consequences for how to evaluate it.
* (-) I believe Sect. 2.2 to misrepresent the state of the art: (a) a number of models that always produce syntactically valid code exist (e.g., https://arxiv.org/abs/1401.0514, https://proceedings.mlr.press/v48/bielik16.html, https://arxiv.org/abs/1611.01855, ...), but these have been supplanted by approaches based on large language models, and (b) most LLMs have no problem generating syntactically valid code. The only evidence for the claim that this is an important issue is provided in the introduction, in the form of a citation of Sequencer, claiming "that up to 70% of codes generated by these models can be non-compilable". However, this is taken from a very specific experiment in the Sequencer paper, which is from 2019 and uses a two layer LSTM; clearly not an LLM.
* (-) Sect. 4: The results here are very small gains over the CodeT5 baseline (when not losing against it), whose significance cannot be interpreted as no error bars are provided. Given that these experiments are based on sampling from the trained models, error bars are crucial. Finally, the entire section does not discuss the computational cost of the proposed method, and in particular, whether spending equal compute on continued training of the underlying CodeT5 baseline would yield better results.
* (-) Sect. 4.4: The ablations are performed only on the "compilation rate" metric, whose value (for users) is not clear; it is however explicitly built into the reward function, and so it remains unclear how meaningful this is.
* (-) Sect. 4.4: Fig. 3 tells us that using a reward term only based on whether the predicted program compiles _reduces_ the rate at which predicted programs compile. I would expect some analysis of the causes here, rather than just moving to "integrating it with the KL-divergence penalty".
* (-) Overall writing is poor, with many phrases often repeating and cluttering the text (e.g., that PPOCoder uses "compiler feedback" and "matching scores" is stated repeatedly, without going into any details)

---

> ### Author Response · Authors · 2023-05-25
> **Response to Reviewer-aDTf**
>
> We would like to thank the reviewer aDTf for the insightful comments. Below, we address your concerns and provide clarification on specific points.
>
> ---
> > Comment:
> > * The results here are very small gains over the CodeT5 baseline (when not losing against it), whose significance cannot be interpreted as no error bars are provided. Given that these experiments are based on sampling from the trained models, error bars are crucial.
>
> **Response**:
> Thanks for your comment. It appears there's a misunderstanding regarding the performance comparisons.
> We would like to clarify that the primary performance improvement exhibited by PPOCoder is indeed reflected in the compilation rates or unit test case pass rates (pass@k), particularly when unit tests are available. The objective of PPOCoder is to guide the priors towards generating more syntactically and functionally correct programs. Therefore, the key metric for assessing PPOCoder's benefit is the compilation rate or unit test pass rate, which is evaluated when the reward component for the compiler feedback is defined based on Eq. (7) or (8) respectively. The PPCOder's results in paper also confirm this as Table 1 and 2 illustrate considerable compilation rate improvement, while Table 3 and 4 display improvements in functional correctness. In the cases where unit tests are not available and we optimize using the compilation test feedback (Eq. (7)), such as in the code completion and translation experiments, we also provide additional metrics such as Exact Match, Edit Sim, and CodeBLEU to compare PPOCoder+CodeT5 with other baselines. These metrics serve to demonstrate that, in addition to the significant improvement in compilation rates, PPOCoder also achieves comparable or even better Edit Sim/CodeBLEU scores compared to the baselines. **This indicates that the generated code by PPOCoder+CodeT5 maintains its semantic meaning and the surge in compilation rate does not derive from generating arbitrary short and easy-to-compile codes**.  This comparable performance of matching metrics is probbaly due to the KL-divergence penalty in the reward that controls explorations and ensures the policy doesn't deviate too far from the pre-trained priors. We have included clearer explanation about this aspect in the revised version of the paper.
>
> In response to your comment on error bars, we agree with the importance of such statistical measures. However, it is worth noting that their inclusion in the studies of code language models is not common, mainly due to the high computational costs needed for model training replications. We are currently performing multiple runs on the code completion task. We will include results in the revised version of paper later as the corresponding experiments are long to be run during the rebuttal period.
>
> ---
> > Comment:
> > * the entire section does not discuss the computational cost of the proposed method, and in particular, whether spending equal compute on continued training of the underlying CodeT5 baseline would yield better results.
>
> **Response**:
> In response to your comment, we conducted experiment (on Java-Py translation) and allowed the token-matching fine-tuning of CodeT5 to continue for the same duration as the optimization of the PPOCoder model for a task. Results demonstrated that the cross-entropy loss had already converged well, and further extensive token-matching fine-tuning did not provide any additional benefit. Therefore, under the same optimization time, PPOCoder+CodeT5 achieved the outperformance (as reported) to the CodeT5 fine-tuned model. This is expected as fine-tuning of the CodeT5 model with token-level matching objectives (in all cases) is stopped when the learning has converged and no further improvement is observed over an extended period, typically around 100 epochs.

---

> > ### Author Response · Authors · 2023-05-25
> > **Response to Reviewer-aDTf (2)**
> >
> > ---
> > > Comment:
> > > * The ablations are performed only on the "compilation rate" metric, whose value (for users) is not clear; it is however explicitly built into the reward function, and so it remains unclear how meaningful this is.
> >
> > **Response**:
> > We want to clarify that compiler signal is one of the four components included in the PPOCoder's reward function. Compiler signal is a sparse feedback received at the end of generation episode which aims to maximize the compilation success or passing of unit tests (when unit tests are available). Compiler signal is defined in Eq. (7) and (8) of the paper.
> >
> > In response to your comment related to ablation study, we have conducted ablation experiments on program synthesis tasks using the MBPP dataset. These studies explore different reward designs and RL algorithms. The results demonstrate the impact of these variations on the model's performance. For instance, the base CodeT5 model (before RL) achieved a pass@80 rate of 32.4. With the default inclusion of KL-divergence penalty, PPOCoder taking only compiler signal reward (Eq.(8)) increases pass@80 performance to 48.7. Additionally, incorporating syntactic and semantic matching scores into the reward function further enhances the performance to 54.3 and 68.2, respectively. Also, comparing RL objectives, we observed performance improvements to 52.6 for +PG, 62.9 for +PG+VF, and 68.2 for CPI+VF, showing that PPO also offers better performance for this task. More detailed results of these ablation experiments can be found in the Appendix (section D, Figure 6).
> >
> > ---
> > > Comment:
> > > * Fig. 3 tells us that using a reward term only based on whether the predicted program compiles reduces the rate at which predicted programs compile. I would expect some analysis of the causes here, rather than just moving to "integrating it with the KL-divergence penalty".
> >
> > **Response**:
> > It appears there's a misunderstanding regarding the ablation results represented in Fig 3(a). We want to point out that the performance drop in the "+cs" case (second bar) of this figure isn't due to the compiler signal hurting the model, but rather due to a lack of exploration constraint. What we wanted to convey is the necessity of divergence constraint in the RL-based fine-tuning due to the broad exploration space and the high potential of policy deviation from the pre-trained priors (i.e., catastrophic forgetting). The third bar in figure confirms this since upon including the KL-divergence penalty, we see a marked improvement. Notably, results show improvement in the compilation rate of +cs model with KL-divergence penalty compared to the base CodeT5 model without RL optimization (first bar), indicating that the compiler signal isn't detrimental but in fact beneficial when incorporated correctly. In response to your comment, we have taken the following steps to address the concerns raised:
> >
> > *  Fig3(a) has been split into two sub-figures to provide a clearer representation of the results. The first shows the effect of reward guidance components (cs, ast, dfg), assuming the inclusion of divergence penalty by default. The second one illustrates the performance of "+cs" model variants with and without the KL-divergence penalty.
> > *  We also conducted additional ablation experiments (same case study) for the model variants without the "+cs" in reward. Results clearly show that model variants with the compiler signal (+cs+dfg: 0.732, +cs+ast: 0.874,  +cs+dfg+ast: 0.921) achieve better compilation rates compared to model variants without the compiler signal (+dfg: 0.694, +ast: 0.801, +dfg+ast: 0.889), with the default inclusion of the KL-divergence penalty in all models. Results can be found in Appendix (section C, Figure 4).

---

> > > ### Author Response · Authors · 2023-05-25
> > > **Response to Reviewer-aDTf (3)**
> > >
> > > ---
> > > > Comment:
> > > > * Given that paper explicitly discusses Python and PHP: What does "compiler signal" mean here - that the code is successfully parsed by python/php? Is type checking used?
> > >
> > > **Response**:
> > > As reviewer noted, Python and PHP PLs are included in the code translation task for XLCoST dataset. For the compilability of Python code, we employ the `py_compile` module to check the syntax of the Python files. Details for terminal command is provided in Appendix (section A.2). On running it will attempt to compile the specified Python file into bytecode. If the file contains any syntax errors, the compilation process will fail and display the error messages indicating the location and nature of the errors. This is used as proxy for a compilation success/failure. For PHP, we employ the PHP linter with the terminal comman provided in Appendix (section A.2). The linter also checks the syntax of the specified PHP file and reports any syntax errors it encounters. If there are any syntax errors, it will output an error message indicating the location and nature of the error. This is again used as proxy for a compilation success/failure. You can also check the`./compiler/compilers.py` module in the provided code link for the implementation of compilers for differnet PLs (Java, C++, C, C#, Pythn, and PHP).
> > >
> > > ---
> > > > Comment:
> > > > * This section misses very important details, foremost how the DFG is computed (given that this is non-trivial in heap-using programs with aliasing) and how DFG edges are matched (finding a graph isomorphism is after all very much non-trivial). Please explain.
> > >
> > > **Response**:
> > > Thank you for pointing out the need for greater detail in this section. We'll make sure to include the implementation details in the revised version of paper.
> > >
> > > In terms of the Data Flow Graph (DFG) computation, particularly in heap-using programs with aliasing, we utilize a points-to analysis to manage the challenge posed by heap aliasing. The analysis provides us with an abstraction of the memory locations a pointer variable could potentially refer to, aiding in constructing a more accurate DFG. We follow the implementation outlined in [4], using the `tree_sitter` library and then the `DFG` module in which the DFG of all languages are created by the use of `tree_to_vvariable_index` method. More implementation details about DFG creation is provided in Appendix. You can also check the `./CodeBLEU/parser/DFG.py` module in the link provided for code.
> > >
> > > For the DFG edge matching, we follow a very simple mechanism adopted in [4]. After DFG creation for both the generated code candidate and the reference target code, we examine each data-flow in the reference DFG. If a match is found in the candidate DFG, we increment the matching data-flow count and eliminate the current data-flow from the reference DFG. More implementation details about DFG edge matching is provided in Appendix. You can also check the `.r/CodeBLEU/dataflow_match.py` module in the link provided for code.
> > >
> > > [4] Ren et al., CodeBLEU: a Method for Automatic Evaluation of Code Synthesis
> > >
> > > ---
> > > > Comment:
> > > > * The choice of R_{ast} is not explained - why is token similarity not sufficient? Using the AST explicitly violates the author's aim to be language-agnostic, after all. The given intuition ("token missing and data type errors") would apply on the token level as well. Please explain.
> > >
> > > **Response**:
> > > Regarding the sufficiency of token matching, it's important to note that tokens in programming languages have significant dependencies and structures that token-level analysis might miss. The abstract syntax tree (AST) provides an explicit structural view of the code and its dependencies, which aids in producing syntactically correct code. To illustrate this, consider the case of a function invocation. If we only have token-level information, a misplaced or missing parenthesis could be difficult to catch, because each parenthesis is a valid token by itself. However, an AST representation would clearly show that the function call is malformed. Also, while token matching could detect a missing token, it may not necessarily identify incorrect usage or placement. This is where the AST-based reward component comes into play.
> > >
> > > Also, it's important to note that PPOCoder is not a model, it's an RL-based fine-tuning framework that can be incoporated with any code generation model. For example, the framework can be incorporated with the translation model between any pair of languages and just by knowing the target language, it can create the AST with the help of `tree_sitter` module called in the reward function. For more details regarding the implementation of this reward component, check `./CodeBLEU/syntax_match.py` module in the link provided for code.

---

> > > > ### Author Response · Authors · 2023-05-25
> > > > **Response to Reviewer-aDTf (4)**
> > > >
> > > > ---
> > > > > Comment:
> > > > > * p1: "given that the generated code is intended for machine execution as opposed to human comprehension": I fundamentally disagree with this statement. Any code meant to be used more than once is also intented to be read by other humans, which has substantial consequences for how to evaluate it.
> > > > > * I believe Sect. 2.2 to misrepresent the state of the art: (a) a number of models that always produce syntactically valid code exist (e.g., https://arxiv.org/abs/1401.0514, https://proceedings.mlr.press/v48/bielik16.html, https://arxiv.org/abs/1611.01855, ...), but these have been supplanted by approaches based on large language models, and (b) most LLMs have no problem generating syntactically valid code. The only evidence for the claim that this is an important issue is provided in the introduction, in the form of a citation of Sequencer, claiming "that up to 70% of codes generated by these models can be non-compilable". However, this is taken from a very specific experiment in the Sequencer paper, which is from 2019 and uses a two layer LSTM; clearly not an LLM.
> > > > > * p10: "absence of pointers and memory management constructs, which can be a source of errors in languages like C++ and C#." - C# has neither pointers nor explicit memory management, please fix this.
> > > > > * Table 3: Explain what the "All" column represents.
> > > >
> > > > **Response**:
> > > > The answer of above comments are provided in order below.
> > > > * We recognize that code is written for both machine execution and human comprehension, particularly when it is intended for reuse or maintenance. We didn't mean to downplay the importance of human readability. However, we want to clarify that the primary focus of our work and the design of PPOCoder is on computer feedback rather than human feedback. This is because most prior work and current baselines in this area evaluate code predominantly based on syntactic and functional correctness, such as passing unit tests. The evaluation of human interpretation of code is a complex task that would require human studies. We have rephrased this statement to reflect the dual-purpose nature of code and the importance of both machine and human interpretability, while explaining the focus on machine feedback due to current common evaluation practices.
> > > > * We want to point out that the cited statement from Sequencer paper is just to provide an example of cases where code generation can lead to non-compilable codes, however, we acknowledge that this may not be representative of the latest, more capable LLMs. Our intention was mainly to shed light on the necessity for improved reliability in code generation.
> > > > * Thank you for highlighting this. We apologize for the oversight. This was supposed to be C instead of C#. We have fixed it in the revision.
> > > > * The 'All' column represents a weighted average of metric results across different APPS difficulty levels - Intro, Interview, and Competition.
> > > >
> > > >
> > > > ---
> > > > > Comment:
> > > > > * p2: "eight programming languages However," -> Missing "."
> > > > > * p6 (and following): R_{dfg} has ugly spacing after the . Please use  (R_{\mathit{dfg}}).
> > > > > * Eq (7): Please reformat so that the is-end-of-token term is on one line, and the other case on the next one - this is unnecessarily hard to parse.
> > > > > * Sect. 2.3 misattributes work (RLHF was not introduced by InstructGPT, as that paper itself says!)
> > > > > * contains sentences that are not parseable (what is "matching scores between the AST sub-trees and DFG edges" supposed to mean?). Please rework this.
> > > > > * Overall, Sect. 3 spends substantial space on recapping fairly standard RL/PPO bits, and misses many details on how the reward function (the core contribution of the paper!) is actually implemented. Please rearrange this, moving standard components into appendix and expanding on the novel contribution.
> > > > > * Overall writing is poor, with many phrases often repeating and cluttering the text (e.g., that PPOCoder uses "compiler feedback" and "matching scores" is stated repeatedly, without going into any details)
> > > > > * Sect. 4.1: Details such as the size of the used CodeT5 model (60M? 220M? 770M?) are missing. Please add.
> > > >
> > > > **Response**:
> > > > Thank you for the comments. We have carefully considered your suggestions and revised the manuscript accordingly.

---

> > > > > ### Author Response · Authors · 2023-06-09
> > > > > **Follow-up**
> > > > >
> > > > > Dear Reviewer,
> > > > >
> > > > > We would like to check whether you have any questions/concerns? We greatly appreciate and welcome any suggestions on improving our paper if the reviewer believes that it will make it easier for the reader to understand.

---

> > > > > ### Comment · Reviewer_aDTf · 2023-06-14
> > > > > **Review Response (Part 4)**
> > > > >
> > > > > >> I believe Sect. 2.2 to misrepresent the state of the art
> > > > > > We want to point out that the cited statement from Sequencer paper is just to provide an example of cases where code generation can lead to non-compilable codes, however, we acknowledge that this may not be representative of the latest, more capable LLMs.
> > > > >
> > > > > You did not update the paper to remove what you acknowledge as a misrepresentation of the state of the art. Please do this.

---

> > > > > > ### Author Response · Authors · 2023-06-16
> > > > > > **Continuing Response to Reviewer-aDTf (4)**
> > > > > >
> > > > > > ---
> > > > > > > Comment:
> > > > > > > * You did not update the paper to remove what you acknowledge as a misrepresentation of the state of the art. Please do this.
> > > > > >
> > > > > > Thank you for bringing this to our attention. We've removed this sentence to avoid confusion.

---

> > > > ### Comment · Reviewer_aDTf · 2023-06-14
> > > > **Reviewer Response (Part 3)**
> > > >
> > > > Thank you for including the additional requested details in the Appendix. As pointed out in my initial review, I believe these should be included in the main text of the paper, which can easily be shortened by moving far more standard definitions from Sect. 3.1 into the appendix.
> > > >
> > > > > Regarding the sufficiency of token matching, it's important to note that tokens in programming languages have significant dependencies and structures that token-level analysis might miss.
> > > >
> > > > While it's true that an AST (or a dataflow graph) can provide additional information, your submission does not actually provide any evidence that this information can be used by the model. To do this, you'd need to include an ablation that only uses a token matching method. I believe that the current "differences between ASTs can be affected by syntactic issues such as token missing and data type errors" should be expanded by concrete examples of what this additional machinery yields. [At the very least, you should fix that sentence, which is not grammatical - "... can be affected by syntactic issues such as missing tokens or data type errors" would be better]

---

> > > > > ### Author Response · Authors · 2023-06-16
> > > > > **Continuing Response to Reviewer-aDTf (3)**
> > > > >
> > > > > We appreciate your feedback and suggestions. To address your points:
> > > > > * At your request, we've moved some details of the implementation from Appendix to the main paper. While we recognize your rationale for shortening section 3.1, we believe the additional information, particularly the methodological details such as definitions and notations, serves to facilitate a deeper understanding of the method for readers. Considering that TMLR does not impose a strict page limit, we prefer to provide comprehensive information to ensure clarity. We will be happy to make changes if deemed necessary.
> > > > > * We have provided different ablation studies to demonstrate the benefits of AST-based and DFG-based reward componenets over the base fine-tuned CodeT5 model. Specifically, the results, located in the Appendix (section D, Figure 4(b)), represent the performance of all model variants without the compiler signal (+cs), i.e., with only the AST-based feedback (+ast), only DFG-based feedback (+dfg), and their integration (+asdt+dfg). In this figure, the first bar is actually referring to a CodeT5 model which is already fine-tuned on the translation task with the token-level matching loss (cross entropy loss). When compared to this baseline, models incorporating the syntactic/semantic matching feedback (+ast/+dfg/+ast+dfg) exhibit enhanced performance, indicating that these mechanisms offer value beyond token-level matching optimization. This evidence suggests that the abstract feedback based on AST and DFG do provide additional and useful information that contributes to the model's performance.
> > > > > * Lastly, we've added more examples to the sentence you pointed out for better clarity.

---

> > > ### Comment · Reviewer_aDTf · 2023-06-14
> > > **Reviewer Response (Part 2)**
> > >
> > > >> The ablations are performed only on the "compilation rate" metric, whose value (for users) is not clear; it is however explicitly built into the reward function, and so it remains unclear how meaningful this is.
> > > >
> > > > Response: We want to clarify that compiler signal is one of the four components included in the PPOCoder's reward function.
> > >
> > > Again, my point is (here and at the base of several of my other comments) is that you have defined a metric not otherwise covered by the literature (compilation rate), integrated this explicitly into your model training procedure and show improvements on it. However, you have not shown that this is an _important_ property to have for usefulness of the models.
> > >
> > > The reductio ad absurdum here would be to propose a wuzzlebuzzleness metric of "programs containing the variable name `wuzzlebuzzle`" and use a reward for programs that contain it - I'm sure the resulting model would produce more programs with the name `wuzzlebuzzle`, but that would clearly not be important.
> > >
> > > Now, compilation is clearly a more relevant property and I think we can all agree that this is the space of reasonable things (unlike wuzzlebuzzleness), but what I feel is missing in your submission is more precise work towards showing the importance of this metric. The fact that stark increases in compilability do not seem to correlate with changes in existing metrics indicates that maybe this property is not all that meaningful.
> > >
> > > > In response to your comment related to ablation study, we have conducted ablation experiments on program synthesis tasks using the MBPP dataset. These studies explore different reward designs and RL algorithms.
> > >
> > > These new experiments in Appendix D go a substantial step towards proving that compilability is a relevant metric. I would encourage you to consider including a scatter plot showing MBPP Pass@k (for whatever k) vs. Compilation Rate. Hopefully, this should show that a higher compilation rate correlates with a higher pass rate?

---

> > > > ### Author Response · Authors · 2023-06-16
> > > > **Continuing Response to Reviewer-aDTf (2)**
> > > >
> > > > Thank you for your feedback and insightful comments. We understand your concern regarding the importance of compilation rate. We want to clarify that passing unit tests is indeed a more comprehensive metric, as it evaluates functional correctness, while compilation rate only checks for syntactic correctness. However, to incorporate unit tests in PPOCoder training or evaluation, the extensive and diverse set of unit tests should be available for each input-output pair of data samples.
> > > > Such unit tests are often not available for widely used datasets in different code generation tasks. This unavailability led us to define the compiler signal based on Eq. (7) or (8), allowing us to use unit tests when available (demonstrated in the program synthesis task for the APPS dataset; Eq. (7)) and to assess compilability when unit tests are not available (Eq.(8)). We would like to emphasize that compiler signal is not the only reward component we optimize for. The reward also includes other components that has shown to guide the model towards generating more functionally correct codes, even when unit tests are not available and hence, aren't considered in the compiler signal (defined based on Eq. (8)) for model optimization.
> > > >
> > > > We hope that the following points help to better clarify the importance of compilation rate:
> > > >
> > > > *  The results from the code translation evaluation on the TransCoder benchmark (provided in the Appendix section C, Table 8) show that when we optimize the code translation CodeT5 model on the XLCoST dataset (where unit tests are not available; compiler signal is based on Eq. (8)), and test it on TransCoder (which has unit tests for some functions), we see improvement in the computation accuracy (passing unit tests) in addition to the compilation error rate. This demonstrates that compiler signal combined with other reward components guide the model towards better functional correctness.
> > > > * Thank you for suggesting the idea of including this scatter plot. As per your advice, we have now included an MBPP pass@1 vs. compilation rate scatter plot in the Appendix section D, Figure 7 (we chose k=1 to better display the difference in compilation rate among various model variants). We want to emphasize that in this task, the program synthesis CodeT5 model is optimized on the APPS dataset (where unit tests are available; compiler signal is based on Eq. (7)), and then tested on the MBPP benchmark for which unit tests are also available. Results in Figure 7 show that the pass@1 improves by integrating different reward terms, and the higher pass@1 corrrelates with higher compilation rate.

---

> > ### Comment · Reviewer_aDTf · 2023-06-14
> > **Reviewer Response (Part 1)**
> >
> > Thank you for your extensive response. I apologise for the late reply - I put this off until the deadline and promptly fell ill, throwing my plans into disarray.
> >
> > > In response to your comment on error bars, [...] We will include results in the revised version of paper later as the corresponding experiments are long to be run during the rebuttal period.
> >
> > Can you report more on these by now?

---

> > > ### Author Response · Authors · 2023-06-16
> > > **Continuing Response to Reviewer-aDTf (1)**
> > >
> > > ---
> > > > Comment:
> > > > * Can you report more on these by now?
> > >
> > > Results are provided in the Appendix (section F, Table 9) of the new revision.

---

### Review · Reviewer_nRFW · 2023-05-10

**Summary Of Contributions:**

This work applies PPO to program synthesis. Specifically, it introduces an optimization goal that balances generating programs that compile with imitating the AST and DFG structure of a reference program in order to provide more fine-grained guidance, showing that only optimizing for compilability tends to yield poor results. The results, based on fine-tuning a pretrained CodeT5 model, suggests that the combined model is particularly capable at generating programs that compile, which yields modest benefits along other metrics.


**Audience:**

Yes

**Broader Impact Concerns:**

The work does not include a broader impact statement and I did not note concerns that would warrant adding one.

**Claims And Evidence:**

No

**Requested Changes:**

Considering the weaknesses stated above, the work should position its contributions more conservatively, improve its evaluation, and include more ablations. Primarily, it needs to highlight that the method it proposes involves significant supervision on the program structure at a fairly low level (esp. matching AST subtrees) rather than optimizing for compilability directly. It should also enrich its set of ablations to (among others, see the above) ensure that simple direct supervision (training to replicate the precise AST and DFG) would not work better for this task, given the apparent negative impact of including compilability reported in Fig. 3(a).

Secondly, it should expand the set of baselines compared in at least one place (adding CodeGen to the APPS comparison). It should either undo or note as a potentially significant caveat the choice to fine-tuning PPOCoder on top of its closest competitor (CodeT5) after fine-tuning the latter. Undoing here means fine-tuning PPOCoder on the originally pretrained CodeT5 so it is compared under the exact same conditions as the CodeT5 model being fine-tuned for each task.

Finally, it should articulate more clearly that the performance improvement primarily manifests as a gain in compilation rates, not necessarily in other downstream task specific metrics. This is particularly true in Tab. 1, where the exact match scores are virtually identical with CodeT5 despite having trained longer on this dataset, and Tab. 2, where the global average CodeBLEU rate of CodeT5 is less than 0.1 points below that of PPOCoder, again despite using less training time.

**Other/minor suggested changes:**

- Please correct the citation style in "Authors of (Chen et al., 2021c)" to "Chen et al. (2021c)"
- Adjust this statement: "Unlike text generation, code generation requires not only syntactic but also functional correctness" -- syntactic correctness is not normally the main objective in text generation, and while functional correctness is not defined in that space, factuality (arguably the closest parallel) certainly is.
- Please elaborate on the motivation behind, and ideally provide experimental evidence for, the specific choice of reward values in Eq. 8, in particular the use of -0.3 and -0.6.
- Please justify the choice for using pass@80 specifically in Table 4


**Strengths And Weaknesses:**

**Strengths:**

+ Presents the first results of applying PPO to program synthesis, highlighting among others that optimizing for compilabiltiy alone does not tend to yield strong results, perhaps due to the sparsity of this signal.
+ Results are generally positive, primarily in terms of generating (far) more compilable programs than baselines, and sometimes by improving results on other metrics, such as pass rate on the APPS benchmark.

**Weaknesses:**

- The approach introduces what amounts to strong supervision on the AST and DFG structure of the generated programs relative to a single target. This significantly undermines the point of using reinforcement learning by imposing the need for a supervised dataset and biasing the training signal towards replicating low-level attributes of its training samples that are only very weakly tied to the real objective of compilation. As such, contrary to the statement in the abstract, the resulting model appears to benefit substantially from supervision on the desired program structure. The work couples this supervision based on AST and DFG matching with the goal of achieving "syntactic and functional correctness", which further highlights the disconnect: the ground-truth programs are functionally and syntactically correct, but that does not imply that the model should learn to generate programs that have largely similar ASTs and DFGs in order to realize these attributes. One of the key motivations for using the pass@K metric is that valid solutions to program synthesis challenges can be implemented very differently. The work also includes no comparisons with some natural alternatives, including supervised approaches that ignore the compilability score (which seems to hurt the model, Fig. 3(a)) or hybrid ones that involve direct supervision on the AST and DFG while using PPO to optimize for compilability. Finally, the choice of DFG and AST scoring functions seem rather ad-hoc. For instance, using only the target function size as the denominator may heavily incentivize PPOCoder to generate very large functions, to maximize overlap. Another ablation that is missing, given the apparent harm of including the compiler feedback just by itself, is one similar to Fig. 3(a) in which all components are added one by one but without the "+cs" part.
- The evaluation involves some questionable decisions. A prominent example is that PPOCoder was fine-tuned on top of an already fine-tuned CodeT5, seemingly in all conditions. That is, CodeT5 is first fine-tuned on datasets such as CodeSearchNet, and only then is PPOCoder fine-tuned on top of this model. This leads to a skewed comparisons, wherein PPOCoder has consistently been trained on the dataset for longer than its main competitor. Another issue is the absence of code-specific large language models in Tab. 3, such as CodeGen. Even the 2B checkpoint would be a much more reasonable baseline than GPT-Neo, which was mostly not trained on code. The results in this table are also somewhat perplexing, suggesting that both CodeRL and PPOCoder widely outperform far larger baselines such as AlphaCode and Codex. I suppose this may be in part due to the use of fine-tuning for some models while Codex and GPT-3 are only prompted (Appendix A.3).
- The focus on compilability makes the contribution relatively narrow. The resulting tool often achieves high compilation rates but comparatively much more modest performance improvements on other metrics. Compilability is also not a particularly impactful optimization target; models like CodeT5 tend to generate compilable outputs at high enough rates (in this work, mostly more than 50%) that sampling could help generate a large set of compilable outputs (which is straightforward since compilability is easy to confirm). A much more interesting objective is yielding functionally correct code, as measured by, e.g., passing unit tests (and certainly not by matching DFGs).

---

> ### Author Response · Authors · 2023-05-25
> **Response to Reviewer-nRFW**
>
> We would like to thank the reviewer nRFW for the insightful and valuable comments. Below, we address your concerns and provide clarification on specific points.
>
> ---
> > Comment:
> > * The work includes no comparisons with some natural alternatives, including supervised approaches that ignore the compilability score (which seems to hurt the model, Fig. 3(a)) or hybrid ones that involve direct supervision on the AST and DFG while using PPO to optimize for compilability.
> > * Another ablation that is missing, given the apparent harm of including the compiler feedback just by itself, is one similar to Fig. 3(a) in which all components are added one by one but without the "+cs" part.
> > * It should also enrich its set of ablations to (among others, see the above) ensure that simple direct supervision (training to replicate the precise AST and DFG) would not work better for this task, given the apparent negative impact of including compilability reported in Fig. 3(a).
> >
>
> **Response**:
> Thank you for the insightful comment. It appears there's a misunderstanding regarding the ablation results represented in Fig 3(a). We want to point out that the performance drop in the "+cs" case (second bar) of this figure isn't due to the compiler signal hurting the model, but rather due to a lack of exploration constraint. What we wanted to convey is the necessity of divergence constraint in the RL-based fine-tuning due to the broad exploration space and the high potential of policy deviation from the pre-trained priors (i.e., catastrophic forgetting). The third bar in figure confirms this since upon including the KL-divergence penalty, we see a marked improvement. Notably, results show improvement in the compilation rate of +cs model with KL-divergence penalty compared to the base CodeT5 model without RL optimization (first bar), indicating that the compiler signal isn't detrimental but in fact beneficial when incorporated correctly. In response to your comment, we have taken the following steps to address the concerns raised:
>
> *  Fig3(a) has been split into two sub-figures to provide a clearer representation of the results. The first shows the effect of reward guidance components (cs, ast, dfg), assuming the inclusion of divergence penalty by default. The second one illustrates the performance of "+cs" model variants with and without the KL-divergence penalty. Fig3(a) and (c) show these two sub-figures in the revised version of the manuscript.
> *  Following reviewer's suggestion, we conducted additional ablation experiments (same case study) for the model variants without the "+cs" in reward. Results clearly show that model variants with the compiler signal (+cs+dfg: 0.732, +cs+ast: 0.874,  +cs+dfg+ast: 0.921) achieve better compilation rates compared to model variants without the compiler signal (+dfg: 0.674, +ast: 0.801, +dfg+ast: 0.879), with the default inclusion of the KL-divergence penalty in all models. Results can be found in Appendix (section D, Figure 4).
> *  We also want to point out that our model employs the syntactic and semantic matching score as abstract, non-differentiable feedback for optimizing the generation of executable syntactic and semantic structures. As we don't decode tree/sub-trees/data-flows, it isn't feasible to directly optimize for structural matching using supervised differentiable objectives. Altering the model architecture to enable such optimization would require significant changes and isn't in accordance with current pre-trained PL models.

---

> > ### Author Response · Authors · 2023-05-25
> > **Response to Reviewer-nRFW (2)**
> >
> > ---
> > > Comment:
> > > * The evaluation involves some questionable decisions. A prominent example is that PPOCoder was fine-tuned on top of an already fine-tuned CodeT5, seemingly in all conditions. That is, CodeT5 is first fine-tuned on datasets such as CodeSearchNet, and only then is PPOCoder fine-tuned on top of this model. This leads to a skewed comparisons, wherein PPOCoder has consistently been trained on the dataset for longer than its main competitor.
> > > * It should either undo or note as a potentially significant caveat the choice to fine-tuning PPOCoder on top of its closest competitor (CodeT5) after fine-tuning the latter. Undoing here means fine-tuning PPOCoder on the originally pretrained CodeT5 so it is compared under the exact same conditions as the CodeT5 model being fine-tuned for each task.
> > >
> >
> > **Response**:
> > Thank you for your insightful comment. Regarding the first point, you correctly highlighted that PPOCoder was optimized on top of the CodeT5 model fine-tuned for each task. This choice was made because CodeT5 is a self-supervised model that requires a pre-train and fine-tune paradigm for generating code in various tasks, as described in the original CodeT5 paper. Pre-training CodeT5 on a large code corpus allows it to learn general representations of code, while the subsequent fine-tuning stage on task-specific datasets improves its performance on code-related tasks. Without this fine-tuning stage, the pre-trained CodeT5 priors lack a sufficient understanding of the generation distributions for each code-related task. PPOCoder builds on top of these fine-tuned priors, which already have a good understanding of the generation distributions for each task, and guides them towards more executable code generations. However, when the quality of generations is already poor for a specific code-related task, PPOCoder cannot effectively optimize for more syntactically or functionally correct generations. Therefore, it is not feasible to undo the fine-tuning and run PPOCoder on top of CodeT5 priors that are not fine-tuned, as the task-specific generations would not be of good quality.
> >
> > Regarding the comparisons, it's important to note that the fine-tuning of the CodeT5 model with token-level matching objectives (in all cases) is stopped when the learning has converged and no further improvement is observed over an extended period, typically around 100 epochs. To confirm this, we previously conducted additional experiment (on Java-Py translation) and allowed the token-matching fine-tuning of CodeT5 to continue for the same duration as the optimization of the PPOCoder model for a task. Results demonstrate that the cross-entropy loss had already converged well, and further extensive token-matching fine-tuning did not provide any additional benefit. Therefore, under the same optimization time, PPOCoder+CodeT5 achieved the outperformance (as reported) to the CodeT5 fine-tuned model. It is crucial to note that the benefit of PPOCoder does not arise from excessive optimization, but rather from the new optimization objective and strategy employed, which effectively guides the model towards more executable code generations. The added value of PPOCoder lies in its ability to optimize code generation beyond what can be achieved by fine-tuning alone.

---

> > > ### Author Response · Authors · 2023-05-25
> > > **Response to Reviewer-nRFW (3)**
> > >
> > > ---
> > > > Comment:
> > > > * it should articulate more clearly that the performance improvement primarily manifests as a gain in compilation rates, not necessarily in other downstream task specific metrics. This is particularly true in Tab. 1, where the exact match scores are virtually identical with CodeT5 despite having trained longer on this dataset, and Tab. 2, where the global average CodeBLEU rate of CodeT5 is less than 0.1 points below that of PPOCoder, again despite using less training time.
> > >
> > > **Response**:
> > > Thank you for highlighting this point. We included a clearer explanation about this in the revised version of the manuscript.
> > > In response to the comment, we would like to clarify that the primary performance improvement exhibited by PPOCoder is indeed reflected in the compilation rates or unit test case pass rates (pass@k), particularly when unit tests are available. The objective of PPOCoder is to guide the priors towards generating more syntactically and functionally correct programs. Therefore, the key metric for assessing PPOCoder's benefit is the compilation rate or unit test pass rate, which is evaluated when the reward component for the compiler feedback is defined based on Eq. (7) or (8) respectively. In the cases where unit tests are not available and we optimize using the compilation test feedback (Eq. (7)), such as in the code completion and translation experiments, we also provide additional metrics such as Exact Match, Edit Sim, and CodeBLEU to compare PPOCoder+CodeT5 with other baselines. These metrics serve to demonstrate that, in addition to the significant improvement in compilation rates, PPOCoder also achieves comparable or even better Edit Sim/CodeBLEU scores compared to the baselines. **This indicates that the generated code by PPOCoder+CodeT5 maintains its semantic meaning and the surge in compilation rate does not derive from generating arbitrary short and easy-to-compile codes**.  This comparable performance of matching metrics is probbaly due to the KL-divergence penalty in the reward that controls explorations and ensures the policy doesn't deviate too far from the pre-trained priors.
> > >
> > > ---
> > > > Comment:
> > > > * The approach introduces what amounts to strong supervision on the AST and DFG structure of the generated programs relative to a single target. This significantly undermines the point of using reinforcement learning by imposing the need for a supervised dataset and biasing the training signal towards replicating low-level attributes of its training samples that are only very weakly tied to the real objective of compilation. As such, contrary to the statement in the abstract, the resulting model appears to benefit substantially from supervision on the desired program structure. The work couples this supervision based on AST and DFG matching with the goal of achieving "syntactic and functional correctness", which further highlights the disconnect: the ground-truth programs are functionally and syntactically correct, but that does not imply that the model should learn to generate programs that have largely similar ASTs and DFGs in order to realize these attributes. One of the key motivations for using the pass@K metric is that valid solutions to program synthesis challenges can be implemented very differently.
> > > > * Primarily, it needs to highlight that the method it proposes involves significant supervision on the program structure at a fairly low level (esp. matching AST subtrees) rather than optimizing for compilability directly.
> > >
> > > **Response**:
> > > Thank you for the insightful comment.
> > > In response to your comment, firstly, it's important to note that in PPOCoder, we do not directly decode sub-trees or data-flows, and therefore, we are not directly optimizing for structural low-level matching with the correct target (i.e., we can't differnetiate over low-level sub-tree structures). Our approach primarily utilizes syntactic and semantic matching scores integrated with compiler signal as abstract and non-differentiable feedback for the entire generated sequence. This means that we are not explicitly biasing the training signal towards replicating low-level attributes of the training samples. However, we acknowledge that by incorporating abstract structural supervision in the reward function, the model may indirectly learn important low-level attributes of correct targets that are strongly correlated with achieving the desired execution outcomes. Also, we acknowledge your point about the pass@K metric, which highlights that multiple valid solutions can exist for a given synthesis problem. While we agree with this idea, our experiments have shown that encouraging similarity in sutructural matchings of training data increases the likelihood of generating functionally correct code for the test data.
> > >
> > > We appreciate your suggestion and we will ensure that the revised version of the paper provides clearer explanations on this aspect.

---

> > > > ### Author Response · Authors · 2023-05-25
> > > > **Response to Reviewer-nRFW (4)**
> > > >
> > > > ---
> > > > > Comment:
> > > > > * The focus on compilability makes the contribution relatively narrow. The resulting tool often achieves high compilation rates but comparatively much more modest performance improvements on other metrics. Compilability is also not a particularly impactful optimization target; models like CodeT5 tend to generate compilable outputs at high enough rates (in this work, mostly more than 50%) that sampling could help generate a large set of compilable outputs (which is straightforward since compilability is easy to confirm). A much more interesting objective is yielding functionally correct code, as measured by, e.g., passing unit tests (and certainly not by matching DFGs).
> > > >
> > > > **Response**:
> > > > Thank you for the comment. PPOCoder is designed to optimize the execution (i.e., syntactic and functional correctness), so, the considerable compilation rate improvement and modest performance improvement on other matching metrics such as CodeBLEU and Edit Sim is expected. We acknowlegde that compilability may not be the best metric to evaluate quality of generated codes and unit test passing is a better choice. To incorporate unit tests in PPOCoder training or evaluation, the extensive and diverse set of unit tests should be available for each input-output pair of data samples.
> > > > However, such unit tests are often not available for widely used datasets in different code generation tasks. This is the reason that we define the compiler feedback based on Eq. (7) or (8), allowing us to consider unit tests when available (as demonstrated in the program synthesis task for the APPS dataset) or to assess compilability when unit tests are not available.
> > > >
> > > > In response to your comment, we have conducted additional experiments on the TransCoder benchmark dataset for code translation task. This dataset consists of 852 parallel functions in C++, Java, and Python PLs, along with some unit tests for certain functions to assess the accuracy of the generated translations. However, the number of test functions with unit tests in TransCoder (446 for C++, 481 for Java, and 463 for Python) is insufficient for RL-based model fine-tuning, given that XLCoST has over 5000 training functions for these languages. Therefore, we utilize TransCoder only as an evaluation benchmark for models trained on XLCoST. Also, since XLCoST and TransCoder both source from the GeeksforGeeks (GFG) website, they have some overlap. We will assess the performance only on the filtered TransCoder test functions that do not coincide with the XLCoST training data. Upon examination, we found that approximately 30% of TransCoder test functions overlap with XLCoST.
> > > > The detailed statistics for different PLs are presented in Appendix (section C, Table 7). Results on TransCoder, summarized below, show that PPOCoder+CodeT5 significantly improves the computational accuracy of CodeT5. Upon examining the failed examples, we also found that PPOCoder considerably reduces compilation errors, as one could expect, demonstrating the efficacy of involving compiler signal in the model fine-tuning. Comprehensive experiment results, covering metrics like computational accuracy, compilation errors, runtime errors, failed unit tests, and timeouts for different language translations are provided in Appendix (section C, Table 8).
> > > >
> > > >
> > > > | Language Pair | CodeT5 Computation Accuracy | PPOCoder+CodeT5 Computation Accuracy | CodeT5 Compilation Error | PPOCoder+CodeT5 Compilation Error |
> > > > |--------------|-------------------------------|---------------------------------------|-------------------------|---------------------------------|
> > > > | C++ → Java   | 39.9                          | 72.1                                  | 42.2                    | 17.4                            |
> > > > | C++ → Py     | 36.8                          | 57.1                                  | 25.2                    | 1.1                             |
> > > > | Java → C++   | 45.2                          | 65.6                                  | 47.9                    | 19.7                            |
> > > > | Java → Py    | 37.4                          | 52.1                                  | 35.3                    | 0.2                             |
> > > > | Py → C++     | 26.1                          | 39.0                                  | 59.9                    | 33.5                            |
> > > > | Py → Java    | 15.9                          | 51.1                                  | 48.3                    | 7.8                             |

---

> > > > > ### Author Response · Authors · 2023-05-25
> > > > > **Response to Reviewer-nRFW (5)**
> > > > >
> > > > > ---
> > > > > > Comment:
> > > > > > * Another issue is the absence of code-specific large language models in Tab. 3, such as CodeGen. Even the 2B checkpoint would be a much more reasonable baseline than GPT-Neo, which was mostly not trained on code. The results in this table are also somewhat perplexing, suggesting that both CodeRL and PPOCoder widely outperform far larger baselines such as AlphaCode and Codex. I suppose this may be in part due to the use of fine-tuning for some models while Codex and GPT-3 are only prompted (Appendix A.3).
> > > > > > * it should expand the set of baselines compared in at least one place (adding CodeGen to the APPS comparison).
> > > > >
> > > > > **Response**:
> > > > > We appreciate your suggestion and agree that comparison to code-specific models like CodeGen is important. Refering to the original CodeGen paper [1] (Table 11), the pass@100 score over MBPP for CodeGen-Multi shows varying performances (46.3, 65.3, 67.9, and 70.02) for model sizes 350M, 2.7B, 6.1B, and 16.1B. The performance improves when Python monolingual pre-training is utilized, the respective scores for the same model sizes increase to 63.00, 74.24, 76.81, and 80.09. In contrast, our PPOCoder+CodeT5 with a 770M model size achieves pass@80 of 68.2, outperforming CodeGen-Multi of sizes 350M, 2.7B, and 6.1B, highlighting competitive performance despite a smaller model size (compared to 2.7B and 6.1B models), and smaller evaluation budget (80 vs. 100 samples). We acknowledge that the largest multi-lingual CodeGen model (CodeGen-Multi-16.1B), and larger mono-lingual CodeGen models (CodeGen-Mono-2.7B, CodeGen-Mono-6.1B, and CoddeGen-Mono-16.1B) surpass our model's performance. However, we believe that comparing our model to CodeGen models, which are 4-20 times larger, may not be entirely fair due to the considerable model size difference and consequently, resources required for fine-tuning. One of the main motivations of PPOCoder is its potential to enhance the competitiveness of smaller models. We believe that this can be really helpful since obtaining execution and structure alignment signal is considerably cheaper than training a model that’s significantly larger.
> > > > >
> > > > > For better comparison, we also looked into the pass@1 performance. The pass@1 performance of PPOCoder+CodeT5 (26.1) surpasses that of all CodeGen-Multi models (7.4, 18.0, 18.3, and 20.9) of varying sizes. Similar to pass@100 results, PPOCoder+CodeT5 pass@1 is better than CodeGen-Mono-350M with pass@1 14.5 and worse than larger CodeGen-Mono models of 2.7B, 6.1B, and 16.1B size with pass@1 of 27.3, 32.4, and 35.2, respectively.
> > > > >
> > > > > In response to your comment, we have included a performance vs. model size pareto comparison between PPOCoder+CodeT5 and the suggested larger models such as CodeGen and PaLM with various size in the Appendix (section E, Figure 7), highlighting the performance-resource trade-off.
> > > > >
> > > > > [1] Nijkamp et al., CodeGen: An Open Large Language Model for Code with Multi-turn Program Synthesis, ICLR 2023
> > > > >
> > > > > ---
> > > > > > Comment:
> > > > > > * the choice of DFG and AST scoring functions seem rather ad-hoc. For instance, using only the target function size as the denominator may heavily incentivize PPOCoder to generate very large functions, to maximize overlap.
> > > > >
> > > > > **Response**:
> > > > > Thank you for the insightful comment. We want to clarify that the denominator in matching scores remains constant as it is based on the correct known target of each sample rather than the generations, which serves as a normalization factor. On the other hand, the numerator depends on the generations and represents the cardinality of overlap between the generated code and the correct target. By maximizing the reward, our goal is to maximize the overlap between the generations and the correct target. During our analysis of the codes generated by PPOCoder, we did not observe a significant trend of generating excessively long functions. This is likely due to the fact that the matching scores are not the only components of the reward function. We also incorporate compiler feedback, which aims to maximize the compilation success or passing of unit tests. This helps to guide the model in generating code that is both concise and functionally correct. Additionally, the KL-divergence penalty in the reward function prevents the model from diverging too far from the initial policy, which helps to discourage the generation of long functions. It is important to note that PPOCoder is also trained with a maximum length constraint for generated code, as outlined in the Appendix (section A).

---

> > > > > > ### Author Response · Authors · 2023-05-25
> > > > > > **Response to Reviewer-nRFW (6)**
> > > > > >
> > > > > > ---
> > > > > > > Comment:
> > > > > > > * Please correct the citation style in "Authors of (Chen et al., 2021c)" to "Chen et al. (2021c)"
> > > > > > > * Adjust this statement: "Unlike text generation, code generation requires not only syntactic but also functional correctness" -- syntactic correctness is not normally the main objective in text generation, and while functional correctness is not defined in that space, factuality (arguably the closest parallel) certainly is.
> > > > > >
> > > > > > **Response**:
> > > > > > Thank you for your comments.
> > > > > > We have carefully considered your suggestions and revised the manuscript accordingly.
> > > > > > Regarding the citation style, we are using the TMLR citation template. We checked other TMLR papers and it seems that the current style is aligned with the existing ones.
> > > > > >
> > > > > > ---
> > > > > > > Comment:
> > > > > > > * Please elaborate on the motivation behind, and ideally provide experimental evidence for, the specific choice of reward values in Eq. 8, in particular the use of -0.3 and -0.6.
> > > > > > > * Please justify the choice for using pass@80 specifically in Table 4
> > > > > >
> > > > > > **Response**:
> > > > > > Thank you for the comments. In response to your comments:
> > > > > > * We use a reward range of -1 to 1 to encompass various unit test outcomes, as suggested by [2].
> > > > > > * We use pass@80 in Table 4 to align with the original paper [3] and ensure consistency with other baselines' reported results.
> > > > > >
> > > > > > [2] Le et al., CodeRL: Mastering Code Generation through Pretrained Models and Deep Reinforcement Learning.
> > > > > >
> > > > > > [3] Austin et al., Program Synthesis with Large Language Models.

---

> > > > > > ### Comment · Reviewer_nRFW · 2023-05-25
> > > > > >
> > > > > > _(edited to combine my comments into one post)_
> > > > > >
> > > > > > I want to thank the authors for providing a thorough response and substantial updates to the paper. They strengthen and clarify the work's contributions. I especially appreciate the new & improved breakdown of the results in Fig. 3 and the adding pass@1 comparisons. A few final notes:
> > > > > >
> > > > > > - While I understand the authors' point that there are few datasets with unit tests that allow for evaluating (let alone fine-tuning) for correctness, I maintain that the paper places far too much emphasis on "functional correctness" given this limitation. Since the PPO optimizations center heavily on compilability and structural imitation, I think the use of this term should be both toned down (it appears seven times in just the abstract and introduction) and that disclaimers should be added earlier in the work to note that no optimization for test passing is included.
> > > > > > - Thanks for adding the thorough analysis of pass@1 scores vs. model sizes in App. E & Fig. 7. This is quite helpful. I would suggest also adding a data point for a plain CodeT5 fine-tuned in the same way, for completeness.
> > > > > > - I agree that the work should not need to outperform CodeGen-16B to be competitive, but my comment around GPT-Neo was more focused on the strangeness of its inclusion in general. Tab. 3 contains quite a few non-code models that should be replaced with code-specific models, such as the CodeGen family as you have done in the additional analyses. I would suggest simply dropping GPT-2 and GPT-Neo from there and inserting CodeGen (mono & multi, 350M and 2B would suffice).
> > > > > > - I recognize that the point of the evaluations is to show that PPOCoder's gains in compilability to do not reduce its performance on other metrics, even if they do not boost it, but I think it is important that the work articulates the corresponding limitation: PPOCoder requires substantial training effort beyond CodeT5 and does not always boost performance on metrics other than compilability by a significant margin (as in Tab. 2). Given that performance on downstream tasks is also sometimes substantially _increased_ (as with TransCoder), it does not hurt the work to acknowledge the downside of incurring the added cost when that is not the case.

---

> > > > > > > ### Author Response · Authors · 2023-05-26
> > > > > > > **Response to Reviewer-nRFW (7)**
> > > > > > >
> > > > > > > We greatly appreciate your insightful comments that really helped to improve our paper's quality. We are glad that our revisions and response have addressed your concerns. We will incorporate your suggestions and ensure that the revised version of the paper addresses the points you raised. Responding to your final comments:
> > > > > > >
> > > > > > > * Thank you for bringing this to our attention. We'll make sure to tone down the usage of this term in the revision. Also, we would like to clarify that while we're not optimizing for test passing in the code completion and code translation tasks, we are including it as a reward component (as defined in Eq.(8)) in the program synthesis task on APPS, where sufficient unit tests are available. We'll add disclaimers to make this distinction clear.
> > > > > > > * We're glad that you found the pass@1 vs. model size analysis helpful. We've added the fine-tuned CodeT5 model to Figure 7 (Appendix, section E) as reviewer suggested.
> > > > > > > * We understand your point about GPT-Neo and GPT-2. The inclusion of these models was inspired by their use in the original Codex [5] and APPS [6] papers, which explicitely reported results for these models fine-tuned on the APPS benchmark (Table 2 in [5] and Table 2 in [6]). We are adopting these results for our comparison. In these papers, authors have mentioned that results are reported from GPT-2 model pretrained on GitHub codes and GPT-Neo that has been pretrained on Pile which also includes GitHub codes. As for the inclusion of CodeGen models in Table 3, we want to add that CodeGen results haven't been reported for this dataset in the original paper. They have reported results on MBPP benchmark and we have included them in the analysis (App. Figure 7). We've tried evaluating pre-trained CodeGen models on the APPS benchmark, however, they seem to perform poor. The authors of CodeGen only reported a pass@5 of 3.84 for CodeGen-16B-Mono (MBPP's best-performing CodeGen model) on Intro APPS problems in their ICLR 2023 openreivew response. This is worse than Codex's 9.65 and other fine-tuned models in Table 3. It's intriguing that pre-trained CodeGen performs very well on MBPP but underperforms on APPS, possibly due to APPS's complexity of problems. For a fair comparison in Table 3, we believe it's best to compare fine-tuned models against CodeGen models also fine-tuned on APPS. However, fine-tuning CodeGen models on APPS for us among problems with different difficulty levels would be time-consuming for the limited response period. We will be happy to make changes if deemed necessary.
> > > > > > > * We appreciate your insight regarding the limitations of PPOCoder.  We'll highlight them in the paper.
> > > > > > >
> > > > > > > [5] Chen et al., Evaluating Large Language Models Trained on Code
> > > > > > >
> > > > > > > [6] Hendrycks et al., Measuring Coding Challenge Competence With APPS

---

> > > > > > > > ### Comment · Reviewer_nRFW · 2023-05-26
> > > > > > > >
> > > > > > > > I thank the authors for their comprehensive response and for addressing the concerns in the paper, where appropriate. I have no further concerns.

---

### Review · Reviewer_F6jH · 2023-05-10

**Summary Of Contributions:**

This paper proposes PPOCoder, which uses the PPO RL algorithm for code generation. They also design some reward functions based on the compiler feedback, and use AST and dataflow graphs to calculate syntactic and semantic matching scores between the model predictions and the ground truth programs. They evaluate their approach based on the CodeT5 model, and the experiments demonstrate good performance on several code generation and translation tasks.

**Audience:**

Yes

**Broader Impact Concerns:**

No broader impact concern.

**Claims And Evidence:**

Yes

**Requested Changes:**

1. For code translation, please evaluate on benchmarks with unit tests, such as Transcoder and HumanEval-X.

2. Add ablation studies of different reward design and RL algorithms on program synthesis tasks.

3. Report MBPP pass@1 accuracies.

4. Add comparison and discussion of experimental results using SOTA pretrained language models for code, such as gpt-3.5-turbo.

**Strengths And Weaknesses:**

Strengths:
Using better RL algorithm and reward functions that are more tailored to code generation is beneficial for improving the performance of language models for code.

Weaknesses:

In general, I think the novelty of this work is very limited. The approach is mostly a combination of existing techniques, including PPO, compiler feedback, and using AST and dataflow graphs to provide more training signals. Despite that it shows better performance than CodeRL on several benchmarks, the improvement is not significant in many cases.

On the other hand, some experiments are not convincing, and the setup should be improved.

1. For code translation, I don't think CodeBLEU score is a convincing evaluation metric. It is better to evaluate on benchmarks with unit tests, such as Transcoder [1] and HumanEval-X [2], and report execution accuracies.

2. I would like to see ablation studies of different reward design and RL algorithms on program synthesis tasks.

3. The experimental results are not compared to the SOTA models. Specifically, I feel that the reported accuracies on MBPP are pretty low. For example, according to the Google PaLM paper [3], pass@80 on MBPP at least already reached >84%.

[1] Lachaux et al., Unsupervised Translation of Programming Languages, NeurIPS 2020.

[2] Zheng et al., CodeGeeX: A Pre-Trained Model for Code Generation with Multilingual Evaluations on HumanEval-X.

[3] Chowdhery et al., PaLM: Scaling Language Modeling with Pathways.

---

> ### Author Response · Authors · 2023-05-25
> **Response to Reviewer-F6jH**
>
> We would like to thank the reviewer F6jH for the insightful and valuable comments. Below, we address your concerns and provide clarification on specific points.
>
> ---
> > Comment:
> > * For code translation, I don't think CodeBLEU score is a convincing evaluation metric. It is better to evaluate on benchmarks with unit tests, such as Transcoder [1] and HumanEval-X [2], and report execution accuracies.
>
> **Response**:
> We appreciate your constructive feedback.
> In response to your comment, we have conducted additional experiments on the TransCoder benchmark dataset for code translation task. This dataset consists of 852 parallel functions in C++, Java, and Python PLs, along with some unit tests for certain functions to assess the accuracy of the generated translations. However, the number of test functions with unit tests in TransCoder (446 for C++, 481 for Java, and 463 for Python) is insufficient for RL-based model fine-tuning, given that XLCoST has over 5000 training functions for these languages. Therefore, we utilize TransCoder only as an evaluation benchmark for models trained on XLCoST. Also, since XLCoST and TransCoder both source from the GeeksforGeeks (GFG) website, they have some overlap. We will assess the performance only on the filtered TransCoder test functions that do not coincide with the XLCoST training data. Upon examination, we found that approximately 30% of TransCoder test functions overlap with XLCoST.
> The detailed statistics for different PLs are presented in Appendix (section C, Table 7). Results on TransCoder, summarized below, show that PPOCoder+CodeT5 significantly improves the computational accuracy of CodeT5. Upon examining the failed examples, we also found that PPOCoder considerably reduces compilation errors, as one could expect, demonstrating the efficacy of involving compiler signal in the model fine-tuning. Comprehensive experiment results, covering metrics like computational accuracy, compilation errors, runtime errors, failed unit tests, and timeouts for different language translations are provided in Appendix (section C, Table 8).
>
>
> | Language Pair | CodeT5 Computation Accuracy | PPOCoder+CodeT5 Computation Accuracy | CodeT5 Compilation Error | PPOCoder+CodeT5 Compilation Error |
> |--------------|-------------------------------|---------------------------------------|-------------------------|---------------------------------|
> | C++ → Java   | 39.9                          | 72.1                                  | 42.2                    | 17.4                            |
> | C++ → Py     | 36.8                          | 57.1                                  | 25.2                    | 1.1                             |
> | Java → C++   | 45.2                          | 65.6                                  | 47.9                    | 19.7                            |
> | Java → Py    | 37.4                          | 52.1                                  | 35.3                    | 0.2                             |
> | Py → C++     | 26.1                          | 39.0                                  | 59.9                    | 33.5                            |
> | Py → Java    | 15.9                          | 51.1                                  | 48.3                    | 7.8                             |
>
> ---
> > Comment:
> > * I would like to see ablation studies of different reward design and RL algorithms on program synthesis tasks.
>
> **Response**:
> Thank you for your suggestion. We have conducted ablation studies on program synthesis tasks using the MBPP dataset. These studies explore different reward designs and RL algorithms. The results demonstrate the impact of these variations on the model's performance. For instance, the base CodeT5 model (before RL) achieved a pass@80 rate of 32.4. With the default inclusion of KL-divergence penalty, PPOCoder taking only compiler signal (Eq.(8)) as reward increases pass@80 performance to 48.7. Additionaly, incorporating syntactic and semantic matching scores into the reward function further enhances the performance to 54.3 and 68.2, respectively. Also, comparing RL objectives, we observed performance improvements to 52.6 for +PG, 62.9 for +PG+VF, and 68.2 for CPI+VF, showing that PPO also offers better performance for this task. More detailed results of these ablation experiments can be found in the Appendix (section D, Figure 6).

---

> > ### Author Response · Authors · 2023-05-25
> > **Response to Reviewer-F6jH (2)**
> >
> > ---
> > > Comment:
> > > * The experimental results are not compared to the SOTA models. Specifically, I feel that the reported accuracies on MBPP are pretty low. For example, according to the Google PaLM paper [3], pass@80 on MBPP at least already reached >84%.
> > > * Report MBPP pass@1 accuracies.
> >
> > **Response**:
> > We appreciate your valuable comment. Referring to the PaLM original paper [1] (Table 12 and Figure 12), pass@80 over MBPP for pre-trained PaLM recorded 35.7, 63.2, and 75.0 for model sizes of 8B, 62B, and 540B respectively. Following fine-tuning, these numbers improve to 60.1, 72.3, and 80.8 for the same model sizes, under the label PaLM-Coder. In our paper (Table 4), we evaluated the zero-shot performance of RL-based models on the MBPP dataset, not the fine-tuned versions. Within this context, our PPOCoder+CodeT5 with a 770M model size outperforms pass@80 scores of PaLM-8B and PaLM-62B, indicating competitive performance despite smaller size. Although PaLM-540B exhibits superior performance on MBPP, we consider a direct comparison with our PPOCoder+CodeT5 potentially misleading due to substantial differences in model sizes and fine-tuning resources. PaLM's top-performing model has 540B parameters, a size vastly greater than PPOCoder+CodeT5's 770M parameters. Yet, our model outperforms larger models, such as PaLM-8B, PaLM-62B, and GPT3-137B. One of the main motivations of PPOCoder is its potential to enhance the competitiveness of smaller models. We believe that this can be really helpful since obtaining execution and structure alignment signal is considerably cheaper than training a model that’s 1000 times larger.
> >
> > The pass@1 performance of PPCoder+CodeT5 with 770M model size is 26.1 which seems to be better than 5.0 and 21.4 pass@1 for PaLM-8B and PaLM-62B. Similar to pass@80 results, PPOCoder+CodeT5 pass@1 is worse than 36.8 pass@1 of largest PaLM model with size 540B.
> >
> > In response to your comment, we have included a performance vs. model size pareto comparison between PPOCoder+CodeT5 and the suggested larger models such as CodeGen and PaLM with various size in the Appendix (section E, Figure 7), highlighting the performance-resource trade-off.

---

> > > ### Author Response · Authors · 2023-06-09
> > > **Follow-up**
> > >
> > > Dear Reviewer,
> > >
> > > We would like to check whether you have any questions/concerns? We greatly appreciate and welcome any suggestions on improving our paper if the reviewer believes that it will make it easier for the reader to understand.

---

> > > > ### Comment · Reviewer_F6jH · 2023-06-11
> > > >
> > > > Thanks for adding new experiments! One more comment I have is that GPT models (GPT-Neo, GPT-2, and GPT-3) and PaLM models (excluding PaLM-Coder) are not sufficiently trained on code, thus claiming that your smaller model outperforms them on coding tasks is not very meaningful. Meanwhile, if you want to claim that PPOCoder is generally helpful for improving the performance of small models, it is helpful to evaluate on more models instead of only CodeT5.

---

> > > > > ### Author Response · Authors · 2023-06-11
> > > > > **Response to Reviewer-F6jH (3)**
> > > > >
> > > > > We appreciate your comment and are glad that our revisions have addressed your previous concerns. We understand your point about GPT-Neo, GPT-2, and PaLM models. It's important to first note that these models are used as baselines in the program synthesis task which is NL2Code. The inclusion of these GPT-based models was inspired by their use in the original Codex [5] and APPS [6] papers, which explicitly reported results for these models fine-tuned on the APPS benchmark (Table 2 in [5] and Table 2 in [6]). We are adopting these results for our comparison. In these papers, authors have mentioned that results are reported from GPT-2 model pretrained on GitHub codes and GPT-Neo that has been pretrained on Pile which also includes GitHub codes. PaLM models also observed code data during pre-training, making them viable candidates for program synthesis tasks (as stated in the original paper).
> > > > >
> > > > > We agree that comparison to code-specific models is also important. We have indeed included different versions of CodeGen models and PaLM-Coder in the performance-resource comparison on MBPP provided in Appendix (section E, Figure 7). Referring to the original CodeGen paper [1] (Table 11), the pass@100 score over MBPP for CodeGen-Multi shows varying performances (46.3, 65.3, 67.9, and 70.02) for model sizes 350M, 2.7B, 6.1B, and 16.1B. The performance improves when Python monolingual pre-training is utilized, the respective scores for the same model sizes increase to 63.00, 74.24, 76.81, and 80.09. In contrast, our PPOCoder+CodeT5 with a 770M model size achieves pass@80 of 68.2, outperforming CodeGen-Multi of sizes 350M, 2.7B, and 6.1B, highlighting competitive performance despite a smaller model size (compared to 2.7B and 6.1B models), and smaller evaluation budget (80 vs. 100 samples). We acknowledge that the largest multi-lingual CodeGen model (CodeGen-Multi-16.1B), and larger mono-lingual CodeGen models (CodeGen-Mono-2.7B, CodeGen-Mono-6.1B, and CoddeGen-Mono-16.1B) surpass our model's performance. However, we believe that comparing our model to CodeGen models, which are 4-20 times larger, may not be entirely fair due to the considerable model size difference and consequently, resources required for fine-tuning. For better comparison, we also looked into the pass@1 performance. The pass@1 performance of PPOCoder+CodeT5 (26.1) surpasses that of all CodeGen-Multi models (7.4, 18.0, 18.3, and 20.9) of varying sizes. Similar to pass@100 results, PPOCoder+CodeT5 pass@1 is better than CodeGen-Mono-350M with pass@1 14.5 and worse than larger CodeGen-Mono models of 2.7B, 6.1B, and 16.1B size with pass@1 of 27.3, 32.4, and 35.2, respectively. Figure 7 in the Appendix show details of these comparisons.
> > > > >
> > > > > It's true that integrating PPOCOder with more code models in addition to CodeT5 would offer a more comprehensive evaluation. However, conducting such experiments with PPOCOder having new backbone models across different tasks and datasets would require a considerable amount of time. Notably, CodeRL was also only tested with CodeT5 backbone. We will be happy to make changes if deemed necessary.
> > > > >
> > > > > [1] Nijkamp et al., CodeGen: An Open Large Language Model for Code with Multi-turn Program Synthesis, ICLR 2023
> > > > >
> > > > > [5] Chen et al., Evaluating Large Language Models Trained on Code
> > > > >
> > > > > [6] Hendrycks et al., Measuring Coding Challenge Competence With APPS

---

### Decision · Action_Editors · 2023-07-01

**Recommendation:** Accept as is

**Comment:**

The paper isn't perfect: the novelty is limited, and there are a few lingering concerns about the baselines used for comparison. However, the overall idea of augmenting language models of code with RL is promising. The paper also delivers an interesting insight: making an ML
model more likely to generate compilable programs doesn't make it more effective at generating correct programs. Finally, the reviewers
appreciate the additional work that the authors put in after the reviews came in and trust the authors to fix the remaining issues (in particular, the discussion of prior work -- see the reviews for more details). Given all this, I am recommending acceptance. Please make sure to incorporate the feedback in the reviews into the final version.

**Audience:**

The paper is on machine learning for code, an increasingly important subarea of machine learning.

**Claims And Evidence:**

The paper gives a framework for code generation in which a pretrained model of code is fine-tuned using an RL objective derived from a code compiler. The method is evaluated on three code generation tasks (NL2code, code translation, and code completion) across several languages and shown to outperform existing language models.